# A Performance-Driven Benchmark for Feature Selection in Tabular Deep Learning

**Valeriia Cherepanova**
University of Maryland

**Roman Levin**[*]
University of Washington

**Gowthami Somepalli**
University of Maryland

**Jonas Geiping**
University of Maryland

**C. Bayan Bruss**
Capital One

**Andrew Gordon Wilson**
New York University

**Tom Goldstein**
University of Maryland

**Micah Goldblum**
New York University

## Abstract

Academic tabular benchmarks often contain small sets of curated features. In contrast, data scientists typically collect as many features as possible into their datasets, and even engineer new features from existing ones. To prevent overfitting in subsequent downstream modeling, practitioners commonly use automated *feature selection* methods that identify a reduced subset of informative features. Existing benchmarks for tabular feature selection consider classical downstream models, toy synthetic datasets, or do not evaluate feature selectors on the basis of downstream performance. Motivated by the increasing popularity of tabular deep learning, we construct a challenging feature selection benchmark evaluated on downstream neural networks including transformers, using real datasets and multiple methods for generating extraneous features. We also propose Deep Lasso – an input-gradient-based analogue of Lasso for neural networks that outperforms classical feature selection methods on challenging problems such as selecting from corrupted or second-order features.

## 1 Introduction

Tabular data is ubiquitous across scientific and industrial applications of machine learning. Practitioners often curate tabular datasets by including exhaustive sets of available features or by hand-engineering additional features. Under such a procedure, real-world tabular datasets can quickly accumulate a large volume of features, many of which are not useful for downstream models. Training on such a large number of features, including noisy or uninformative ones, can cause overfitting. To avoid overfitting, practitioners filter out and remove features using automated *feature selection* methods [Guyon and Elisseeff, 2003, Liu and Yu, 2005].

The literature contains a wide body of work proposing or evaluating feature selection approaches that use classical machine learning algorithms in their selection criteria or which select features for training classical machine learning algorithms downstream [Tibshirani, 1996, Kohavi and John, 1997]. Over the past few years, deep learning for tabular data has become competitive and is increasingly adopted by practitioners [Arık and Pfister, 2021, Gorishniy et al., 2021, Somepalli et al., 2021]. While the community has identified that neural networks are especially prone to overfitting on noisy features

---

[*]The substantive contributions of the author to the work described in the paper were done prior to the author joining Amazon.

37th Conference on Neural Information Processing Systems (NeurIPS 2023) Track on Datasets and Benchmarks.

[Grinsztajn et al., 2022], a thorough evaluation of feature selection methods for downstream tabular neural networks is lacking.

To address this absence, we benchmark feature selection methods in tabular deep learning setting by evaluating the selected features via the performance of neural networks trained on them downstream. In addition to popular existing methods, we select features using the attention maps of tabular transformer models [Gorishniy et al., 2021], and we further propose Deep Lasso – an input-gradient-based analogue of Lasso [Tibshirani, 1996] for deep tabular models. Whereas many previous works on feature selection use entirely synthetic datasets [Wah et al., 2018, Passemiers et al., 2023, Bolón-Canedo et al., 2013] or create extraneous features by concatenating random noise onto existing features [Dinh and Ho, 2020, Borisov et al., 2019], we conduct our benchmark on real datasets, and explore three different ways to construct extraneous features: random noise features, corrupted features, and second-order features that serve as a prototypical example of feature engineering.

In our experiments, we find that while many feature selection methods can differentiate between informative features and noise within reason, they may fail under more challenging settings. Notably, Deep Lasso selects features which achieve significantly better downstream neural network performance than previous methods when selecting from corrupted or second-order features. [1]

Our primary contributions are summarized as follows:

- We construct a challenging feature selection benchmark comprising real-world datasets with extraneous uninformative, corrupted, and redundant features.
- We benchmark different feature selection algorithms for deep tabular models, including recent tabular transformer architectures, on the basis of downstream performance.
- We propose a generalization of Lasso for deep neural networks, which leverages input gradients to train models robust to changes in uninformative features. We show that Deep Lasso outperforms other feature selection methods, including tree-based methods, in the most challenging benchmark setups.

## 2  Related Work

In the following section, we provide a brief overview of recent developments in tabular deep learning, and of feature selection in machine learning more broadly.

### 2.1  Tabular Deep Learning

Tabular data is the dominant format of data in real-world machine learning applications. Until recently, these applications were primarily solved using classical decision tree models, such as gradient boosted decision trees (GBDT). However, modern deep tabular neural networks started to bridge the gap to conventional GBDTs, which in turn unlocked new use cases for deep learning in tabular domain. Recent work includes the development of novel tabular architectures, for example based on transformer models [Huang et al., 2020, Gorishniy et al., 2021, Somepalli et al., 2021, Arık and Pfister, 2021], and ensembles of differentiable learners [Popov et al., 2019, Kontschieder et al., 2015, Hazimeh et al., 2020, Badirli et al., 2020], as well as modifications and regularizations for MLP-based architectures [Kadra et al., 2021, Gorishniy et al., 2022]. Other works explore new capabilities that are enabled by tabular deep learning, such as self-supervised pre-training [Ucar et al., 2021, Somepalli et al., 2021, Rubachev et al., 2022, Kossen et al., 2021, Agarwal et al., 2022], transfer learning [Levin et al., 2023, Wang and Sun, 2022, Zhu et al., 2023], few-shot learning [Nam et al., 2023] and data generation [Kotelnikov et al., 2022].

### 2.2  Feature Selection

A cornerstone of applied machine learning is feature selection, where data science practitioners carefully curate and select features for predictive tasks. As a result, there has been considerable interest in automating this process.

---

[1]Our code for benchmark and Deep Lasso is available at `https://github.com/vcherepanova/tabular-feature-selection`

Existing approaches for feature selection can be categorized into three main types: *filter*, *wrapper* and *embedded* methods. Filtering algorithms rank features based on their individual characteristics and relevance to target variables, without considering any specific learning algorithm. Examples of filter methods include univariate statistical tests, variance filters, and mutual information scores. A comprehensive overview of existing filter methods can be found in [Lazar et al., 2012]. Wrapper methods, on the other hand, are algorithm-dependent and involve iteratively re-training a machine learning algorithm on a subset of features to identify the subset that yields the best performance. These include greedy sequential algorithms [Kittler, 1978], recursive feature elimination [Guyon et al., 2002, Huang et al., 2018] as well as evolutionary algorithms [Xue et al., 2015, Siedlecki and Sklansky, 1989, Kennedy and Eberhart, 1995, Gheyas and Smith, 2010]. Embedded methods incorporate the task of feature selection into the training process, allowing the model to learn which features are most relevant while training. Lasso [Tibshirani, 1996] is a classical embedded feature selection algorithm, which has been also applied to deep neural networks in the form of Adaptive Group Lasso [Dinh and Ho, 2020]. Additionally, tree-based algorithms like Random Forests [Breiman, 2001] and Gradient Boosted Decision Trees [Friedman, 2001] employ built-in feature importance measures, enabling automatic feature selection. A few recent works propose specialized neural network architectures with embedded feature selection through knockoff filters [Lu et al., 2018, Zhu et al., 2021], auto-encoders [Balın et al., 2019, Zhu et al., 2021] and specialized gating layers [Lemhadri et al., 2021].

We find wrapper methods generally too computationally expensive to be useful for the deep neural network models we consider in this study, especially when hyperparameter optimization is performed. Therefore, in our study we focus on established filter and embedded approaches, both classical and modern, that can be applied to generic tabular architectures.

### 2.3 Feature Selection Benchmarks

So far, it is unclear which strategy, whether classical or modern, would be optimal for feature selection with deep tabular models. Existing benchmark studies focus primarily on classical downstream models [Bolón-Canedo et al., 2013, Bommert et al., 2020, Bolón-Canedo et al., 2014] or do not optimize for downstream performance at all [Lu et al., 2018]. A few works evaluate feature selection methods on synthetic datasets [Bolón-Canedo et al., 2013, Wah et al., 2018, Passemiers et al., 2023, Sánchez-Maroño et al., 2007] or on domain specific datasets such as high-dimensional genomics data containing small number of samples [Bommert et al., 2020, Bolón-Canedo et al., 2014], malware detection data [Darshan and Jaidhar, 2018] and text classification [Darshan and Jaidhar, 2018, Forman et al., 2003]. Passemiers et al. [2023] evaluates neural network interpretability methods in a feature selection setup on synthetic datasets, using a small MLP model, and finds them to be less effective than classical feature selection methods such as random forest importance.

In contrast, in this work we provide an extensive benchmark of feature selection methods for modern deep tabular models. We construct our feature selection benchmark on large-scale tabular datasets with different types of extraneous features and investigate a range of representative feature selectors, both classical and deep learning based. Our benchmark evaluates feature selection methods based on performance of the downstream deep tabular models.

## 3 Experimental Setup

We construct a challenging feature selection benchmark that uses real datasets and includes multiple approaches for the controlled construction of extraneous features. In all cases, we evaluate feature selection methods based on downstream neural network performance, i.e. the practically relevant metric of success for a feature selection procedure. We conduct experiments with both MLP architectures and the recent transformer-based deep tabular FT-Transformer architecture [Gorishniy et al., 2021] as downstream models. Our benchmark comprises 12 datasets with 3 types of additional features. These datasets are collected and adapted based on recent tabular benchmark papers [Gorishniy et al., 2021, 2022, Rubachev et al., 2022] and include ALOI (AL), California Housing (CA), Covertype (CO), Eye Movements (EY), Gesture (GE), Helena (HE), Higgs 98k (HI), House 16K (HO), Jannis (JA), Otto Group Product Classification (OT), Year (YE) and Microsoft (MI). Among these datasets, there are eight classification datasets and four regression datasets. We measure downstream model performance using accuracy for the classification tasks and RMSE for the regression tasks. Additional details concerning these datasets can be found in Appendix B.1.

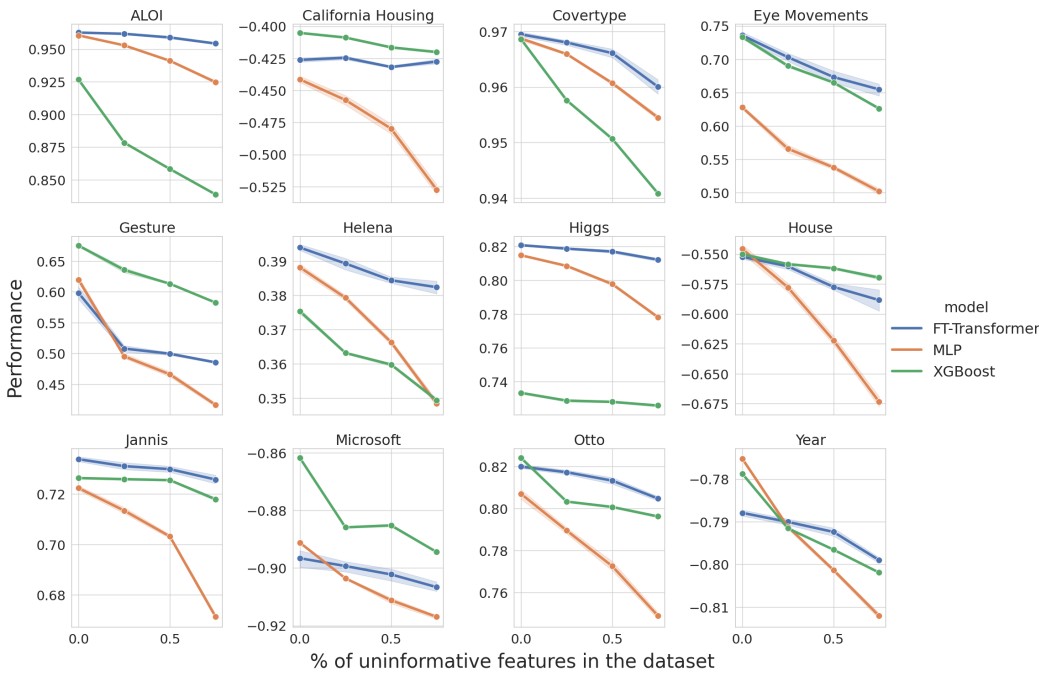

Figure 1: **Performance of FT-Transformer, MLP and XGBoost models when trained on data with random extra features.** The X-axes indicates the percentage of uninformative features in the dataset and the Y-axes depict accuracy for classification problems and negative RMSE for regression datasets, so higher values always indicate better performance. Overall, we find MLP models to be more susceptible to noise than either XGBoost or FT-Transformer models.

For each benchmarking experiment, we perform extensive hyperparameter tuning for both feature selection algorithms and downstream models with respect to the downstream model performance using the Bayesian hyperparameter optimization engine Optuna [Akiba et al., 2019]. We select the best hyperparameters based on validation metrics and report test metrics computed over 10 random model initializations (seeds). Details concerning final hyperparameters can also be found in the Appendix sections D, C.3.

In the following section, we present a motivating experiment. In Section 5 we discuss our benchmark design in detail. Section 6 presents experimental benchmark results using feature selection methods.

## 4 Are Deep Tabular Models More Susceptible to Noise than GBDT?

Recent contributions to the ongoing competition between tabular neural networks and gradient boosted decision trees (GBDT) have found that neural networks are more susceptible to noise than GBDT on small to medium datasets (up to 10,000 samples) [Grinsztajn et al., 2022]. We scale this experiment to larger datasets and showcase it as a motivating example for feature selection methods and benchmarks specific to deep tabular models.

We explore the influence of uninformative features on tabular neural networks and assess the performance of MLP and FT-Transformer models on datasets containing varying numbers of uninformative Gaussian noise features. For reference, we also include a GBDT model into our comparison as implemented in the popular XGBoost package [Chen and Guestrin, 2016]. Figure 1 illustrates the relationship between the performance of the three models and the proportion of uninformative features in these datasets. Similarly to Grinsztajn et al. [2022], we observe that the MLP architecture, on average, exhibits more overfitting to uninformative features compared to XGBoost, motivating the need for careful feature selection with tabular neural networks. Interestingly, as seen from the slope of the blue and green curves in Figure 1, the FT-Transformer model is roughly as robust to noisy features as the XGBoost model. The fact that the performance of the FT-Transformer model is not as

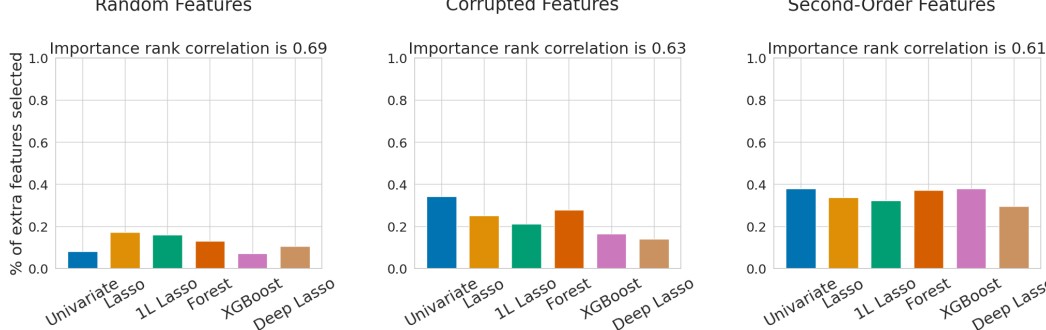

Figure 2: **Percent of random, corrupted, and second-order features selected by different feature selection algorithms.** *Importance rank correlation* refers to pair-wise feature importance Spearman correlation averaged across all feature selection algorithms and datasets. Random features are less often ranked as important compared to corrupted and second-order features, and feature selection algorithms have higher agreement when selecting from random features.

severely affected by noisy features could be attributed to the ability of the transformer architecture to filter out uninformative features through its attention mechanism. Inspired by this observation, we further investigate the effectiveness of utilizing the attention map importance within FT-Transformer Gorishniy et al. [2021] as a feature selection method in our benchmark study.

## 5 Feature Selection Benchmark

It is not common for real-world datasets to contain completely random noise features with no predictive power whatsoever, although engineered features often exhibit varying degrees of redundancy and noise. Nonetheless, feature selection algorithms are often evaluated on datasets containing spurious features generated from Gaussian noise. Not only does this differ significantly from real-world feature selection scenarios, but it also presents a relatively straightforward task for many feature selection algorithms to eliminate these random features. In light of these considerations, we propose an alternative approach to establish a more challenging and realistic feature selection benchmark by introducing three distinct methods for crafting additional features:

**Random Features.** In the simplest scenario, we sample uninformative features from a Gaussian distribution and concatenate them with the original dataset features.

**Corrupted Features.** To simulate a scenario with noisy, yet still relevant features, we sample extraneous features from the original ones and corrupt them with Gaussian noise. In addition, we conduct experiments with Laplace noise corruption and report experimental results in Appendix F.1.

**Second-Order Features.** To simulate a feature engineering scenario with redundant information contained in engineered features, we add second-order features, i.e. products of randomly selected original features.

Note that higher-order features are not spurious and are often used by data scientists precisely because they can contain useful information. Therefore, selecting higher order features instead of original ones may not be a bad thing. Such feature selection algorithms must be evaluated in terms of downstream model performance as we do in the following section.

To gauge the difficulty of the proposed benchmark, we explore how often different feature selection algorithms rank extraneous features among the top-$k$ most important features, where $k$ represents the number of original features in the datasets. From Figure 2, we observe that all methods select fewer random features than they do corrupted or second-order features. Additionally, to quantify the overall agreement between different feature selection methods, we analyze the average pair-wise Spearman correlation between the rankings of features generated by different selection algorithms. Notably, the setup involving random extra features exhibits the highest correlation, indicating that filtering out random features is relatively straightforward and all feature selection algorithms behave similarly. In

contrast, the setup with second-order extra features has the lowest rank correlation implying greater disparity in selection preferences among the algorithms.

We note, that different feature selection algorithms may select similar features, but rank features differently within "important" and "unimportant" groups, which would results in lower rank correlation even in the settings where feature selection is straightforward (i.e. selecting from random features).

# 6 Benchmarking Feature Selection Methods

In this section, we benchmark various feature selection methods. In particular, we consider the following feature selection approaches:

**Univariate Statistical Test**. This classical analysis checks the linear dependence between the predictors and the target variable. It selects features based on the ANOVA F-values for classification problems and univariate linear regression test F-values for regression problems.

**Lasso** uses $L_1$ regularization to encourage sparsity in a linear regression model [Tibshirani, 1996]. After this sparse regression, features are ranked with respect to the magnitudes of their coefficients in the model.

**First-Layer Lasso (1L Lasso)** is an extension of Lasso for MLPs with multiple layers. It applies a Group Lasso penalty to the weights of the first layer parameters:

$$\min_\theta \alpha \mathcal{L}_\theta(X, Y) + (1 - \alpha) \sum_{j=1}^m ||W^{(j)}||_2,$$

where $W^{(j)}$ is the j-th column of weight matrix of the first hidden layer corresponding to the $j$-th feature. Similarly to Lasso, First-Layer Lasso ranks features with respect to their grouped weights in the first layer.

**Adaptive Group Lasso (AGL)** is an extension of the Group Lasso regularization method [Dinh and Ho, 2020]. Similarly to the First-Layer Lasso, it applies a Group Lasso penalty to the weights of the first layer parameters, however each group of coefficients is weighted with an adaptive weight parameter:

$$\min_\theta \alpha \mathcal{L}_\theta(X, Y) + (1 - \alpha) \sum_{j=1}^m \frac{1}{||\hat{W}^{(j)}||_2^\gamma} ||W^{(j)}||_2,$$

where $\hat{W}$ is the Group Lasso estimate of $W$. Adaptive Group Lasso then ranks features with respect to their grouped weights in the first layer.

**LassoNet** is a neural network architecture that incorporates feature selection [Lemhadri et al., 2021]. LassoNet achieves feature sparsity by adding a skip (residual) layer and allowing the features to participate only if their skip-layer connection is active.

**Random Forest (RF)** is a bagging ensemble of decision trees, and it ranks features with respect to their contribution to the ensemble [Breiman, 2001]. In particular, importance is calculated by measuring the decrease in impurity when that feature is used for splitting at each node of the decision trees in the forest.

**XGBoost** is a popular implementation for gradient boosted decision tree [Chen and Guestrin, 2016]. XGBoost computes feature importance as the average gain across all splits in trees where a feature was used.

**Attention Map Importance (AM)** is computed for FT-Transformer model from one forward pass on the validation set. We follow Gorishniy et al. [2021] and calculate feature importance as the average attention map for the `[CLS]` token across all layers, heads, and validation samples.

**Deep Lasso** is our generalization of Lasso to deep tabular models (and, in fact, any differentiable model). Deep Lasso encourages feature gradient sparsity for deep tabular models by applying a Group Lasso penalty to gradients of the loss with respect to input features during training. Intuitively, this makes the model robust to changes in unimportant features. For train data $(X, Y)$ with $n$ samples and $m$ features, the Deep Lasso penalty is given by

Table 1: **Benchmarking feature selection methods for MLP and FT-Transformer downstream models on datasets with random extra features.** We report performance of models trained on features selected by different FS algorithms in terms of accuracy for classification and negative RMSE for regression problems. % refers to percent of extra features in the dataset: either 50% or 75% features are random. Bold font indicates the best numbers dataset-wise and lower rank indicates better overall result.

| % | FS method | AL | CH | CO | EY | GE | HE | HI | HO | JA | MI | OT | YE | rank |
|---|---|---|---|---|---|---|---|---|---|---|---|---|---|---|
| 50 | No FS + MLP | 0.941 | -0.480 | 0.961 | 0.538 | 0.466 | 0.366 | 0.798 | -0.622 | 0.703 | -0.911 | 0.773 | -0.801 | 8.08 |
| 50 | Univariate + MLP | **0.96** | -0.447 | 0.970 | 0.575 | 0.515 | 0.379 | 0.811 | **-0.549** | 0.715 | **-0.891** | **0.808** | -0.776 | 2.66 |
| 50 | Lasso + MLP | 0.949 | -0.454 | 0.969 | 0.547 | 0.458 | 0.380 | 0.812 | -0.599 | 0.715 | -0.907 | 0.805 | -0.787 | 5.91 |
| 50 | 1L Lasso + MLP | 0.952 | -0.451 | 0.969 | 0.564 | 0.474 | 0.375 | 0.811 | -0.568 | 0.715 | -0.897 | 0.796 | **-0.773** | 4.91 |
| 50 | AGL + MLP | 0.958 | -0.512 | 0.969 | **0.578** | 0.473 | **0.386** | 0.810 | -0.557 | 0.718 | -0.898 | 0.799 | -0.778 | 4.33 |
| 50 | LassoNet + MLP | 0.954 | **-0.445** | 0.969 | 0.552 | 0.495 | 0.385 | 0.811 | -0.557 | 0.715 | -0.907 | 0.783 | -0.787 | 5.16 |
| 50 | AM + MLP | 0.953 | **-0.444** | 0.968 | 0.554 | 0.498 | 0.382 | **0.813** | -0.566 | **0.722** | -0.904 | 0.801 | -0.777 | 3.83 |
| 50 | RF + MLP | 0.955 | -0.453 | 0.969 | **0.589** | **0.594** | **0.386** | **0.814** | -0.572 | 0.720 | -0.904 | 0.806 | -0.786 | 2.91 |
| 50 | XGBoost + MLP | 0.956 | **-0.444** | 0.969 | **0.59** | 0.502 | **0.385** | 0.812 | -0.560 | **0.72** | -0.893 | 0.805 | -0.777 | **2.33** |
| 50 | Deep Lasso + MLP | **0.959** | -0.443 | 0.968 | 0.573 | 0.485 | 0.383 | 0.814 | **-0.549** | **0.72** | -0.894 | 0.802 | -0.776 | **2.33** |
| 75 | No FS + MLP | 0.925 | -0.527 | 0.955 | 0.502 | 0.417 | 0.348 | 0.778 | -0.674 | 0.671 | -0.917 | 0.749 | -0.812 | 7.41 |
| 75 | Univariate + MLP | **0.96** | -0.447 | 0.970 | 0.575 | 0.502 | 0.381 | 0.810 | **-0.549** | 0.713 | **-0.89** | 0.806 | -0.776 | 2.50 |
| 75 | Lasso + MLP | 0.959 | -0.454 | 0.967 | 0.543 | 0.491 | 0.381 | 0.811 | -0.612 | 0.716 | -0.907 | 0.802 | -0.789 | 4.33 |
| 75 | 1L Lasso + MLP | 0.957 | **-0.448** | 0.968 | 0.555 | 0.432 | 0.380 | 0.809 | -0.572 | 0.717 | -0.903 | 0.799 | **-0.775** | 4.41 |
| 75 | AGL + MLP | 0.954 | **-0.447** | 0.968 | 0.561 | 0.429 | 0.382 | 0.809 | -0.571 | 0.719 | -0.901 | 0.762 | -0.777 | 4.33 |
| 75 | LassoNet + MLP | 0.958 | -0.452 | 0.966 | 0.528 | 0.475 | 0.383 | 0.809 | **-0.555** | 0.705 | -0.913 | 0.768 | -0.794 | 4.75 |
| 75 | RF + MLP | 0.949 | -0.453 | 0.968 | **0.584** | **0.61** | 0.386 | **0.814** | -0.585 | 0.718 | -0.902 | **0.808** | -0.784 | 2.91 |
| 75 | XGBoost + MLP | 0.958 | -0.451 | 0.969 | 0.576 | 0.583 | 0.382 | 0.810 | -0.568 | **0.72** | -0.892 | 0.804 | **-0.774** | **2.08** |
| 75 | Deep Lasso + MLP | 0.957 | **-0.446** | 0.969 | 0.569 | 0.479 | **0.387** | **0.814** | -0.559 | 0.721 | -0.893 | 0.800 | **-0.774** | 2.33 |
| 50 | No FS + FT | 0.959 | -0.432 | 0.966 | 0.673 | 0.500 | 0.384 | 0.817 | -0.577 | 0.730 | -0.902 | 0.813 | -0.792 | 6.58 |
| 50 | Univariate + FT | **0.963** | -0.424 | 0.970 | 0.700 | 0.519 | 0.389 | 0.819 | **-0.554** | 0.733 | **-0.897** | 0.819 | -0.789 | 2.83 |
| 50 | Lasso + FT | 0.952 | **-0.419** | 0.960 | 0.682 | 0.489 | 0.388 | 0.819 | -0.594 | 0.728 | -0.999 | 0.817 | -0.998 | 6.33 |
| 50 | 1L Lasso + FT | 0.963 | -0.423 | 0.969 | **0.72** | 0.489 | 0.382 | 0.818 | -0.577 | 0.732 | -0.904 | **0.819** | -0.791 | 5.16 |
| 50 | AGL + FT | 0.899 | **-0.42** | 0.969 | 0.701 | 0.480 | 0.393 | **0.822** | -0.586 | 0.733 | -0.915 | 0.814 | -0.832 | 5.25 |
| 50 | LassoNet + FT | **0.963** | -0.426 | 0.970 | 0.670 | 0.505 | 0.392 | 0.818 | -0.559 | 0.733 | -0.904 | 0.808 | -0.791 | 4.83 |
| 50 | AM + FT | 0.962 | -0.425 | 0.968 | 0.657 | 0.505 | 0.389 | 0.820 | **-0.554** | 0.735 | -0.903 | 0.815 | **-0.789** | 3.75 |
| 50 | RF + FT | 0.963 | **-0.42** | 0.969 | **0.718** | **0.591** | **0.395** | 0.821 | -0.558 | **0.737** | -0.900 | **0.82** | -0.791 | **1.75** |
| 50 | XGBoost + FT | **0.963** | **-0.42** | 0.969 | **0.725** | 0.572 | 0.392 | 0.820 | -0.558 | 0.734 | **-0.898** | **0.82** | -0.789 | 1.91 |
| 50 | Deep Lasso + FT | 0.962 | **-0.419** | 0.969 | 0.703 | 0.504 | 0.392 | 0.817 | -0.560 | 0.733 | -0.900 | 0.817 | **-0.788** | 3.66 |

$$\min_{\theta} \alpha \mathcal{L}_\theta(X,Y) + (1-\alpha) \sum_{j=1}^{m} \left\| \frac{\partial \mathcal{L}_\theta(X,Y)}{\partial X^{(j)}} \right\|_2, \tag{1}$$

where, for $j$-th feature (column of $X$), $\frac{\partial \mathcal{L}_\theta(X,Y)}{\partial X^{(j)}} = \left( \frac{\partial \mathcal{L}_\theta(X,Y)}{\partial x_{1j}}, \frac{\partial \mathcal{L}_\theta(X,Y)}{\partial x_{2j}}, \ldots, \frac{\partial \mathcal{L}_\theta(X,Y)}{\partial x_{nj}} \right)$. Once the model is trained with the above regularizer, the corresponding feature importance is provided by

$$\text{importance of the } j\text{-th feature} = \left\| \frac{\partial \mathcal{L}_\theta(X,Y)}{\partial X^{(j)}} \right\|_2. \tag{2}$$

Note that in the linear regression case, the classical Lasso is equivalent to the proposed input-gradient sparsity regularizer applied to model output since in the linear case input gradients are the weights of the linear model. We provide a formal proof of equivalence between Deep Lasso, classical Lasso, and First-Layer Lasso in the linear regression case in Appendix Section E. We also note that Deep Lasso is related to methods used to promote network explainability by leveraging input gradients [Sundararajan et al., 2017, Shrikumar et al., 2017, Smilkov et al., 2017, Levin et al., 2022]. In addition, [Liu, 2021] examines input gradients for feature selection from a Bayesian perspective. This work uses Bayesian Neural Networks (BNNs) along with associated credible intervals to estimate uncertainty surrounding input gradients, choosing features based on the plausibility that their corresponding gradients are zero. On the one hand, this principled work comes with theoretical guarantees, but on the other hand it requires Hamiltonian Monte Carlo, an expensive sampler which does not scale to large datasets and models.

## 6.1 Results

We benchmark feature selection methods for downstream MLP and FT-Transformer models. In the case of the MLP downstream model, we explore scenarios where either 50% or 75% of the features

Table 2: **Benchmarking feature selection methods for MLP and FT-Transformer downstream models on datasets with corrupted extra features.** We report performance of models trained on features selected by different FS algorithms in terms of accuracy for classification and negative RMSE for regression problems. % refers to percent of extra features in the dataset: either 50% or 75% features are corrupted. Bold font indicates the best numbers dataset-wise and lower rank indicates better overall performance.

| % | FS method + Model | AL | CH | CO | EY | GE | HE | HI | HO | JA | MI | OT | YE | rank |
|---|---|---|---|---|---|---|---|---|---|---|---|---|---|---|
| 50 | No FS + MLP | 0.946 | -0.475 | 0.965 | 0.557 | 0.525 | 0.370 | 0.802 | -0.607 | 0.703 | -0.909 | 0.778 | -0.797 | 8.00 |
| 50 | Univariate + MLP | **0.955** | -0.451 | 0.966 | 0.556 | 0.514 | 0.346 | 0.810 | -0.620 | 0.717 | -0.920 | 0.795 | -0.828 | 7.33 |
| 50 | Lasso + MLP | **0.955** | **-0.449** | 0.968 | 0.548 | 0.512 | 0.382 | 0.813 | -0.602 | 0.713 | -0.903 | 0.796 | -0.795 | 5.42 |
| 50 | 1L Lasso + MLP | **0.955** | -0.447 | 0.968 | 0.566 | 0.515 | 0.382 | 0.812 | -0.581 | 0.718 | -0.902 | 0.795 | -0.780 | 4.75 |
| 50 | AGL + MLP | 0.953 | -0.450 | 0.968 | **0.588** | 0.538 | 0.386 | 0.813 | -0.561 | 0.722 | -0.902 | 0.796 | -0.780 | 3.00 |
| 50 | LassoNet + MLP | **0.955** | -0.452 | **0.969** | 0.570 | 0.556 | 0.382 | 0.811 | **-0.551** | 0.719 | -0.905 | 0.795 | -0.777 | 3.83 |
| 50 | AM + MLP | **0.955** | **-0.449** | 0.967 | **0.583** | 0.527 | 0.381 | 0.814 | **-0.555** | 0.722 | -0.905 | 0.797 | -0.780 | 3.58 |
| 50 | RF + MLP | 0.951 | -0.453 | 0.967 | 0.574 | **0.568** | 0.383 | 0.810 | -0.565 | **0.724** | -0.904 | 0.788 | -0.786 | 4.67 |
| 50 | XGBoost + MLP | 0.954 | -0.454 | **0.969** | **0.583** | 0.510 | 0.385 | **0.815** | -0.553 | 0.722 | **-0.892** | **0.803** | -0.779 | 2.67 |
| 50 | Deep Lasso + MLP | **0.955** | -0.447 | 0.968 | 0.577 | 0.525 | **0.388** | **0.815** | -0.567 | 0.721 | -0.895 | 0.801 | **-0.776** | 2.58 |
| 75 | No FS + MLP | 0.921 | -0.516 | 0.956 | 0.518 | 0.503 | 0.356 | 0.788 | -0.632 | 0.686 | -0.913 | 0.762 | -0.808 | 7.58 |
| 75 | Univariate + MLP | 0.955 | -0.569 | 0.941 | 0.510 | 0.495 | 0.347 | 0.742 | -0.620 | 0.686 | -0.921 | 0.779 | -0.838 | 7.50 |
| 75 | Lasso + MLP | 0.948 | -0.454 | 0.963 | **0.565** | 0.490 | 0.373 | 0.810 | -0.593 | 0.717 | -0.903 | 0.795 | -0.791 | 5.00 |
| 75 | 1L Lasso + MLP | 0.955 | -0.444 | 0.967 | 0.549 | 0.495 | 0.380 | 0.811 | -0.576 | **0.715** | -0.903 | 0.797 | -0.779 | 3.25 |
| 75 | AGL + MLP | 0.928 | -0.566 | 0.967 | 0.548 | 0.490 | 0.382 | 0.811 | -0.574 | 0.714 | -0.904 | 0.788 | -0.780 | 4.92 |
| 75 | LassoNet + MLP | 0.947 | -0.452 | **0.969** | 0.539 | **0.533** | 0.383 | 0.805 | -0.572 | 0.708 | -0.908 | 0.791 | -0.785 | 4.33 |
| 75 | RF + MLP | 0.952 | -0.450 | 0.963 | 0.547 | **0.533** | 0.372 | 0.805 | -0.573 | 0.716 | -0.903 | 0.765 | -0.788 | 4.92 |
| 75 | XGBoost + MLP | 0.954 | -0.515 | 0.968 | **0.571** | 0.53 | 0.381 | 0.811 | **-0.571** | **0.721** | **-0.895** | 0.800 | -0.784 | 2.58 |
| 75 | Deep Lasso + MLP | **0.959** | **-0.441** | 0.968 | 0.554 | 0.517 | **0.386** | 0.813 | -0.563 | 0.718 | -0.898 | **0.804** | -0.778 | **1.42** |
| 50 | No FS + FT | 0.960 | -0.430 | 0.967 | 0.686 | 0.576 | 0.386 | 0.818 | -0.574 | 0.731 | -0.901 | 0.809 | -0.793 | 6.08 |
| 50 | Univariate + FT | **0.963** | -0.422 | 0.965 | 0.681 | 0.574 | 0.345 | 0.812 | -0.628 | 0.733 | -0.920 | 0.812 | -0.826 | 6.17 |
| 50 | Lasso + FT | 0.952 | -0.422 | 0.936 | 0.697 | 0.556 | 0.387 | 0.820 | -0.586 | 0.732 | -0.937 | 0.812 | -0.915 | 6.42 |
| 50 | 1L Lasso + FT | 0.962 | **-0.419** | 0.969 | 0.718 | 0.571 | 0.389 | 0.820 | -0.570 | 0.731 | **-0.899** | **0.816** | -0.795 | 3.50 |
| 50 | AGL + FT | 0.906 | -0.426 | 0.969 | 0.697 | 0.591 | **0.392** | 0.820 | -0.552 | **0.735** | -0.914 | **0.816** | -0.830 | 4.08 |
| 50 | LassoNet + FT | **0.962** | -0.426 | **0.97** | 0.679 | 0.578 | **0.393** | 0.814 | -0.572 | **0.736** | -0.903 | 0.813 | **-0.79** | 3.92 |
| 50 | AM + FT | 0.962 | -0.424 | 0.969 | 0.680 | 0.572 | **0.392** | 0.820 | **-0.549** | 0.734 | -0.901 | **0.817** | -0.790 | 3.42 |
| 50 | RF + FT | 0.962 | -0.422 | 0.969 | 0.711 | **0.6** | 0.387 | 0.819 | -0.557 | 0.735 | -0.898 | 0.806 | -0.793 | 3.42 |
| 50 | XGBoost + FT | **0.963** | -0.422 | **0.97** | 0.706 | 0.564 | 0.392 | **0.821** | -0.548 | 0.735 | -0.897 | **0.816** | -0.790 | **2.42** |
| 50 | Deep Lasso + FT | 0.961 | -0.422 | 0.968 | **0.725** | 0.577 | **0.393** | **0.821** | -0.561 | **0.736** | -0.898 | 0.809 | **-0.788** | 2.67 |

in the dataset are extraneous. For the FT-Transformer, we focus on evaluating FS methods solely on datasets with 50% added features. For simplicity, we train the downstream models on the top-k important features determined by the feature selection algorithms, where k corresponds to the original number of features in the datasets. We include more details on our experimental setup in Section C.

We report dataset-wise downstream performance based on the proposed benchmark, as well as overall rank of the methods in Tables 1, 2, and 3 and we include results with standard errors computed across seeds in Tables 16, 17, 18. While we did not discover a universally optimal feature selection method that performed the best across all datasets and extra feature types, we did identify several noteworthy patterns, which we will discuss below.

**Random Features.** In the easy scenario where extraneous features are Gaussian noise, we observe in Table 1 that XGBoost, Random Forest, univariate statistical test and Deep Lasso perform on par for MLP downstream models, while for FT-Transformer downstream models, Random Forest and XGBoost outperform other methods. Conversely, Lasso, 1L Lasso, AGL, LassoNet and Attention Map Importance are less competitive. These findings align with the results depicted in Figure 2, which highlight the high similarity in importance rank between the methods and the greater tendency of Lasso based methods to assign higher ranks to random features. In addition to downstream performance, we report ROC-AUC and precision scores in Tables 11, 12.

**Corrupted Features.** In a more challenging scenario involving corrupted extra features, both Deep Lasso and XGBoost significantly outperform the other feature selection methods. Specifically, Deep Lasso exhibits superior performance for the MLP downstream model, while XGBoost performs slightly better for the FT-Transformer downstream model, see Table 2.

**Second-Order Features.** Finally, in the most challenging scenario of choosing among original and second-order features, Deep Lasso demonstrates a significant performance advantage over the other methods, see Table 3. Interestingly, we discover that the relative rank of Deep Lasso is lower when 75% of all features were generated, indicating that Deep Lasso excels in more challenging feature

Table 3: **Benchmarking feature selection methods for MLP and FT-Transformer downstream models on datasets with second-order extra features.** We report performance of models trained on features selected by different FS algorithms in terms of accuracy for classification and negative RMSE for regression problems. % refers to percent of extra features in the dataset: either 50% or 75% features are second-order. Bold font indicates the best numbers dataset-wise and lower rank indicates better overall result.

| % | FS method | AL | CH | CO | EY | GE | HE | HI | HO | JA | MI | OT | YE | rank |
|---|---|---|---|---|---|---|---|---|---|---|---|---|---|---|
| 50 | No FS + MLP | 0.960 | -0.443 | 0.969 | 0.631 | 0.605 | **0.383** | 0.811 | **-0.549** | 0.719 | -0.891 | 0.800 | -0.786 | 4.50 |
| 50 | Univariate + MLP | **0.961** | -0.439 | 0.959 | 0.584 | 0.582 | 0.357 | 0.817 | -0.614 | 0.724 | -0.902 | 0.798 | -0.810 | 6.58 |
| 50 | Lasso + MLP | 0.955 | -0.443 | 0.966 | 0.608 | 0.590 | 0.366 | 0.816 | -0.564 | 0.724 | -0.891 | **0.806** | -0.783 | 5.33 |
| 50 | 1L Lasso + MLP | 0.959 | -0.445 | 0.969 | 0.634 | 0.571 | 0.380 | **0.815** | -0.565 | 0.728 | **-0.89** | **0.808** | -0.780 | 3.92 |
| 50 | AGL + MLP | 0.961 | -0.443 | **0.953** | 0.637 | 0.594 | 0.383 | 0.807 | -0.565 | 0.730 | **-0.89** | 0.806 | **-0.776** | 3.25 |
| 50 | LassoNet + MLP | 0.959 | -0.442 | 0.969 | **0.641** | 0.611 | 0.379 | 0.816 | -0.595 | 0.724 | -0.893 | 0.797 | -0.784 | 4.50 |
| 50 | AM + MLP | **0.961** | **-0.439** | 0.968 | 0.622 | 0.604 | 0.381 | **0.819** | -0.566 | 0.730 | -0.892 | 0.802 | -0.778 | 3.50 |
| 50 | RF + MLP | 0.958 | **-0.437** | 0.969 | 0.639 | **0.619** | 0.370 | 0.818 | -0.586 | 0.735 | **-0.89** | 0.801 | -0.781 | 3.25 |
| 50 | XGBoost + MLP | 0.870 | **-0.438** | 0.97 | 0.635 | 0.604 | 0.373 | **0.818** | -0.579 | **0.734** | -0.891 | 0.805 | -0.786 | 3.83 |
| 50 | Deep Lasso + MLP | **0.961** | -0.441 | 0.969 | **0.648** | 0.600 | **0.384** | 0.815 | -0.572 | 0.733 | **-0.89** | 0.805 | **-0.776** | **2.67** |
| 75 | No FS + MLP | 0.952 | -0.451 | **0.969** | 0.630 | 0.598 | **0.388** | 0.808 | **-0.542** | 0.717 | -0.900 | 0.792 | -0.792 | 4.92 |
| 75 | Univariate + MLP | 0.960 | -0.530 | 0.488 | 0.553 | 0.531 | 0.352 | 0.812 | -0.608 | 0.720 | -0.908 | 0.785 | -0.820 | 7.25 |
| 75 | Lasso + MLP | **0.96** | **-0.434** | **0.968** | 0.612 | 0.519 | 0.363 | 0.820 | -0.554 | 0.739 | -0.894 | **0.807** | -0.793 | 3.67 |
| 75 | 1L Lasso + MLP | **0.96** | -0.452 | 0.966 | **0.654** | 0.579 | 0.375 | 0.818 | -0.549 | **0.741** | -0.893 | 0.805 | -0.782 | 3.33 |
| 75 | AGL + MLP | 0.958 | -0.438 | **0.968** | 0.647 | 0.601 | 0.384 | 0.819 | **-0.545** | 0.736 | -0.893 | 0.800 | -0.781 | 2.67 |
| 75 | LassoNet + MLP | 0.958 | -0.454 | 0.968 | 0.633 | **0.615** | 0.362 | 0.813 | -0.569 | 0.726 | -0.895 | 0.793 | -0.786 | 4.83 |
| 75 | RF + MLP | 0.956 | -0.445 | 0.968 | 0.627 | 0.566 | 0.339 | 0.819 | -0.615 | 0.728 | -0.892 | 0.794 | -0.789 | 5.08 |
| 75 | XGBoost + MLP | 0.459 | -0.495 | **0.968** | 0.627 | 0.555 | 0.358 | **0.821** | -0.588 | 0.738 | -0.892 | 0.803 | -0.795 | 5.00 |
| 75 | Deep Lasso + MLP | 0.959 | -0.448 | **0.969** | **0.647** | 0.582 | 0.378 | **0.821** | -0.568 | **0.74** | **-0.89** | 0.805 | **-0.78** | **2.17** |
| 50 | No FS + FT | 0.962 | -0.425 | 0.968 | 0.733 | 0.558 | **0.391** | 0.819 | **-0.552** | 0.732 | -0.901 | 0.818 | -0.790 | 4.08 |
| 50 | Univariate + FT | **0.963** | **-0.419** | 0.942 | 0.615 | 0.573 | 0.351 | 0.820 | -0.613 | 0.728 | -0.910 | 0.810 | -0.819 | 6.25 |
| 50 | Lasso + FT | 0.962 | -0.422 | 0.967 | 0.720 | 0.560 | 0.381 | **0.823** | -0.561 | 0.733 | -0.905 | 0.816 | -0.874 | 5.08 |
| 50 | 1L Lasso + FT | **0.964** | -0.421 | 0.970 | **0.748** | **0.604** | **0.391** | 0.812 | -0.557 | 0.736 | -0.896 | 0.817 | -0.795 | 2.83 |
| 50 | AGL + FT | 0.935 | **-0.421** | 0.950 | **0.747** | 0.540 | 0.388 | 0.821 | **-0.557** | 0.730 | -0.907 | **0.821** | -0.821 | 4.16 |
| 50 | LassoNet + FT | **0.963** | -0.423 | 0.970 | 0.729 | **0.612** | 0.378 | 0.817 | -0.592 | 0.733 | -0.901 | 0.808 | -0.792 | 5.00 |
| 50 | AM + FT | 0.963 | **-0.418** | 0.970 | 0.719 | 0.597 | 0.387 | 0.820 | -0.579 | 0.736 | -0.902 | 0.819 | -0.791 | 3.83 |
| 50 | RF + FT | **0.963** | -0.422 | 0.970 | 0.733 | **0.615** | 0.365 | 0.822 | -0.600 | **0.738** | **-0.895** | 0.816 | -0.793 | 3.66 |
| 50 | XGBoost + FT | 0.871 | -0.424 | 0.962 | 0.737 | 0.594 | 0.379 | 0.822 | -0.587 | 0.736 | **-0.896** | 0.812 | -0.792 | 4.08 |
| 50 | Deep Lasso + FT | **0.963** | -0.422 | 0.970 | 0.726 | **0.608** | 0.388 | 0.822 | -0.558 | 0.738 | -0.897 | 0.819 | **-0.789** | **2.41** |

selection problems involving a significant proportion of spurious or redundant features, as observed in both corrupted and second-order extra features.

# 7 Similarity between Feature Selection Methods

In this section we analyze which feature selection algorithms are more similar to each other in terms of feature ranking. In Figure 3, we present a heatmap of pair-wise Spearman correlation between feature rankings of different FS methods averaged across datasets. We focus on the setup involving second-order features, as we discovered the lowest level of agreement between methods within this configuration. It is worth noting that we provide results for methods fine-tuned for both MLP and FT-Transformer models, ensuring a comprehensive comparison.

We find related similarity patterns for both downstream models. In particular, we observe that Random Forest and XGBoost rankings exhibit high correlation, indicating a strong agreement between these both tree-based methods. We find that First-Layer Lasso is highly correlated with Adaptive Group Lasso, which is an extension of First-Layer Lasso. Additionally, Random Forest results are also correlated with results from the Univariate Test. Deep Lasso is most correlated with Lasso-based methods, such as classical Lasso, First-Layer Lasso and Adaptive Group Lasso and with Attention Map Importance.

# 8 Discussion

In this paper, we present a challenging feature selection benchmark for deep tabular models, addressing the need for more realistic scenarios in feature selection evaluation. Our benchmark encompasses real-world datasets with extraneous uninformative, corrupted, and redundant features. Through comprehensive experimentation, we compare various feature selection methods on our proposed benchmark. Our findings reveal that while classical feature selection methods, including tree-based

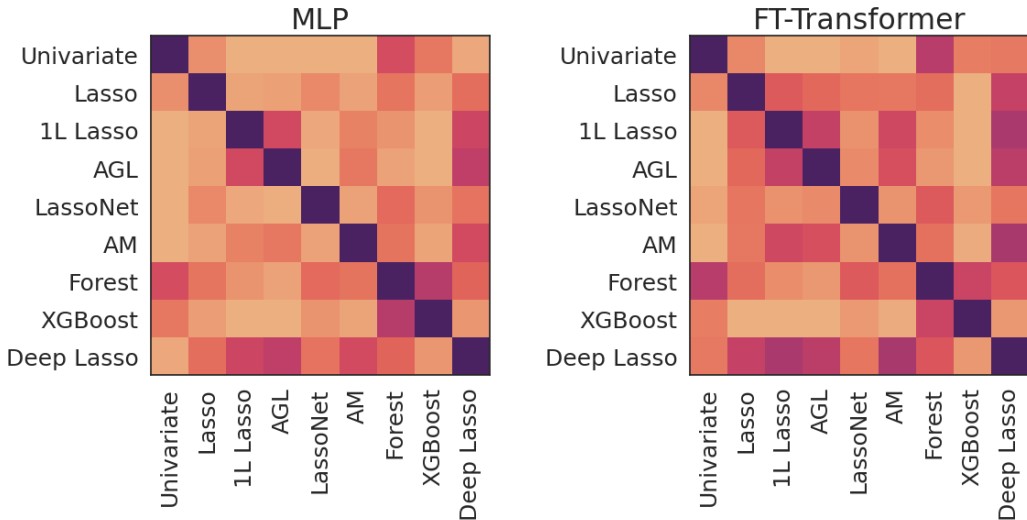

Figure 3: **Similarity between feature selection methods tuned for downstream MLP and FT-Transformer models.** Heatmap depicts pair-wise Spearman correlation of feature importances for the setup with second-order features. Correlations are averaged across the datasets.

algorithms perform competitively in the random and corrupted setups, specialized deep tabular feature selection methods, like the proposed Deep Lasso outperform other methods in selecting from second-order features. This indicates the benefits of neural network inductive biases in feature selection algorithms. Overall, our study contributes a systematic new benchmark with analysis, a new feature selection method, and insights into improving the performance and robustness of deep tabular models. The benchmark code and Deep Lasso implementation are made available to facilitate reproducibility and practical usage.

## Acknowledgements

This work was made possible by the ONR MURI program and the AFOSR MURI program. Commercial support was provided by Capital One Bank, the Amazon Research Award program, and Open Philanthropy. Further support was provided by the National Science Foundation (IIS-2212182), and by the NSF TRAILS Institute (2229885).

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

# A   Limitations

In this section we discuss limitations of our work. We note that Deep Lasso feature selection is more computationally demanding than using GBDT-based feature selection methods, especially when large-scale hyperparameter optimization is performed. In fact, for this work we ran over 80 thousand experiments. Combined with computationally intense methods, this may have resulted in excessive carbon emissions. In addition, while we introduce challenging feature selection tasks and use sizeable datasets, the scale of these datasets is still smaller than truly large-scale industrial tabular data with billions of samples.

# B   Benchmark Detail

Our benchmark comprises of datasets used in [Gorishniy et al., 2021, 2022]. We select datasets with up to 150 numerical features and no categorical features.

## B.1   Datasets

- ALOI [Geusebroek et al., 2005] (AL, image data)
- California Housing [Pace and Barry, 1997] (CH, real estate data)
- Covertype [Blackard and Dean, 1999] (CO, forest characteristics)
- Eye Movements [Salojärvi et al., 2005] (EY, eye movement trajectories)
- Gesture Phase Prediction [Madeo et al., 2013] (GE, gesture phase segmentation)
- Helena [Guyon et al., 2019] (HE, anonymized)
- Higgs 98 [Baldi et al., 2014] (HI, simulated physical particles)
- House 16H [2] (HO, house pricing)
- Jannis [Guyon et al., 2019] (JA, anonymized)
- Microsoft [Qin and Liu, 2013] (MI, search queries)
- Otto [3] (OT, product categories)
- Year [Bertin-Mahieux et al., 2011] (YE, audio features)

Table 4: Dataset Details. Number of classes is reported for classification problems.

|  | AL | CH | CO | EY | GE | HE | HI | HO | JA | MI | OT | YE |
|---|---|---|---|---|---|---|---|---|---|---|---|---|
| # samples | 108000 | 20640 | 581012 | 10936 | 9873 | 65196 | 98050 | 22784 | 83733 | 1200192 | 61878 | 515345 |
| # features | 128 | 8 | 54 | 26 | 32 | 27 | 28 | 16 | 54 | 136 | 93 | 90 |
| # classes | 1000 | - | 7 | 3 | 5 | 100 | 2 | - | 4 | - | 9 | - |

## B.2   Extraneous Features

In the random features setup we draw extraneous features from the Gaussian distribution. After preprocessing the original features with quantiler transformation, both original and extraneous features follow normal distribution. In the corrupted features setup we randomly sample $n$ original features and corrupt them with Gaussian noise with the same standard deviation $x_c = 0.5x + 0.5\epsilon, \epsilon \sim \mathcal{N}(0, \sigma)$. For generating second-order features we randomly sample two subsets of original features and compute their products.

---

[2]https://www.openml.org/search?type=data&sort=runs&id=574
[3]https://www.kaggle.com/c/otto-group-product-classification-challenge/data

## B.3 Data Preprocessing

We follow [Gorishniy et al., 2021] and apply quantiler transformation to normalize numerical features in all datasets, except for Helena and ALOI, where standardization is applied instead. For the regression tasks we also standardize the target variable. Due to limited computational resources, we restrict the number of samples in the Microsoft dataset to 500,000. We reserve 20% of samples for test set and 15% of samples as a validation set for tuning the hyperparameters of feature selection and downstream models.

## B.4 Code

We include the benchmark code as well as instructions for reproducing our results in the supplementary. We will release our code as a GitHub Repository under the MIT License.

# C   Experimental Detail

## C.1 Hardware

We ran our experiments on NVIDIA GeForce RTX 2080 Ti machines. Each hyperparameter tuning experiment took from 1 to 72 hours depending on the model and the dataset size. Overall we ran 756 experiments, each with hyperparameter tuning for 100 trials and 10 subsequent seed runs, which resulted in $\sim 83160$ training runs.

## C.2 Implementation Licenses

For the model implementations we adapt the code from the following publicly available repositories and libraries:

- MLP and FT-Transformer models from RTDL repository [4] under MIT License

- GBDT model from XGBoost library [5] under Apache License

- Random Forest, Lasso and Univariate statistical test models from Scikit-Learn library [6] under BSD 3-Clause License.

## C.3 Training Details

We train all deep tabular models for 200 epochs with early stopping after 20 epochs, meaning that training stops if the validation accuracy does not improve after 20 consecutive epochs. XGBoost models are trained with a maximum of 2000 estimators and 50 early stopping rounds. We utilize the AdamW optimizer and apply a Linear learning rate scheduler after epochs 40 and 80. The batch size for all experiments is set to 512. Other training parameters, including learning rate and weight decay, are determined through hyperparameter tuning.

## C.4 Statistical Significance

We run each experiment with 10 random model initializations (seeds) after we find the optimal hyperparameters for feature selection and downstream models. We report average test metrics in Tables 1 2 3 as well as model ranks. When calculating the average ranks, we consider the statistical significance of performance differences among the models. Specifically, we assign rank 1 to the top-performing models that do not exhibit statistically significant differences. We determine statistical significance with the one-sided Wilcoxon Rank-Sum test Wilcoxon [1945], Mann and Whitney [1947] with $p = 0.05$.

---

[4]`https://github.com/Yura52/tabular-dl-revisiting-models`
[5]`https://xgboost.ai/`
[6]`https://scikit-learn.org/stable/`

# D   Hyperparameter Tuning

We carefully tune the parameters of all models for each experiment using the Bayesian hyperparameter optimization library Optuna[Akiba et al., 2019]. For the feature selection experiments we tune parameters for both feature selection and downstream models simultaneously to optimize for downstream validation accuracy. We run 100 trials to identify the optimal parameters for each experiment. We adapt the hyperparameter search spaces from the original papers with slight modifications, which are provided below.

- Hyperparameters for the MLP model are provided in Table 5
- Hyperparameters for the FT-Transformer model are provided in Table 6
- Hyperparameters for the XGBoost feature selection model are provided in Table 7
- Hyperparameters for the Random Forest feature selection model are provided in Table 8
- Hyperparameters for feature selection regularizers (Lasso, Deep Lasso, First-Layer Lasso, Adaptive Group Lasso) are provided in Table 9

Table 5: Optuna hyperparameter search space for MLP

| Parameter | Search Space |
|---|---|
| Number of layers | $\text{UniformInt}[1, 8]$ |
| Layer size | $\text{UniformInt}[1, 512]$ |
| Dropout | $\{0, \text{Uniform}[0, 0.5]\}$ |
| Learning rate | $\text{LogUniform}[1e-5, 1e-2]$ |
| Weight decay | $\{0, \text{LogUniform}[1e-6, 1e-3]\}$ |

Table 6: Optuna hyperparameter search space and default configuration for FT-Transformer

| Parameter | Search Space | Default |
|---|---|---|
| Number of layers | $\text{UniformInt}[1, 4]$ | 3 |
| Residual dropout | $\{0, \text{Uniform}[0, 0.2]\}$ | 0.0 |
| Attention dropout | $\text{Uniform}[0, 0.5]$ | 0.2 |
| FFN dropout | $\text{Uniform}[0, 0.5]$ | 0.1 |
| FFN factor | $\text{Uniform}[2/3, 8/3]$ | 4/3 |
| Learning rate | $\text{LogUniform}[1e-5, 1e-3]$ | $1e-3$ |
| Weight decay | $\text{LogUniform}[1e-6, 1e-3]$ | $2e-4$ |

Table 7: Optuna hyperparameter search space for XGBoost

| Parameter | Search Space |
|---|---|
| Max depth | $\text{UniformInt}[3, 10]$ |
| Min child weight | $\text{LogUniform}[1e-8, 1e5]$ |
| Subsample | $\text{Uniform}[0.5, 1]$ |
| Learning rate | $\text{LogUniform}[1e-5, 1]$ |
| Col sample by tree | $\text{Uniform}[0.5, 1]$ |
| Gamma | $\{0, \text{LogUniform}[1e-8, 1e2]\}$ |
| Lambda | $\{0, \text{LogUniform}[1e-8, 1e2]\}$ |

Table 8: Optuna hyperparameter search space for Random Forest

| Parameter | Search Space |
|---|---|
| Num estimators | $\text{UniformInt}[10, 2000]$ |
| Max depth | $\text{UniformInt}[3, 10]$ |

Table 9: Optuna hyperparameter search space for Feature Selection regularizers penalty weights

| Feature Selection | Penalty Search Space |
|---|---|
| Lasso | Uniform $[1e-3, 5e-1]$ |
| Deep Lasso | LogUniform$[1e-2, 5e-1]$ |
| First-Layer Lasso | LogUniform$[1e-2, 5e-1]$ |
| Adaptive Group Lasso | LogUniform$[1e-3, 5e-1]$ |

For certain datasets training XGBoost (ALOI dataset) and FT-Transformer (Microsoft, Year, ALOI, Covtype) models can be excessively time-consuming. Due to these time constraints, we do not perform parameter tuning for these particular pairs of models and datasets and instead utilize the default configurations of the models. We also note that the computational complexity of attention mechanisms grows quadratically as the number of features in the dataset increases. Finally, LassoNet implementation performs hyperparameter tuning automatically, therefore we do not use Optuna to tune LassoNet model, and only tune parameters of the downstream models.

## E  Deep Lasso for Linear Regression

Consider the case of linear regression, where the model is

$$f_{w,b}(X) = Xw^T + b.$$

We follow the PyTorch-style notation and $w$ is a row vector with $m$ entries where $m$ is the number of features – columns of $X$. We show that Deep Lasso applied to the model output (rather than loss) is equivalent to regular Lasso and First-Layer Lasso.

*Proof:*
Deep Lasso applied to the model output is given by:

$$\min_\theta \alpha \mathcal{L}_\theta(X, Y) + (1 - \alpha) \sum_{j=1}^m \left\| \frac{\partial f_{w,b}(X)}{\partial X^{(j)}} \right\|_F. \tag{3}$$

Now, for $j$-th feature (column of $X$), we have

$$\frac{\partial f_{w,b}(X)}{\partial X^{(j)}} = \frac{\partial (Xw^T + b)}{\partial X^{(j)}} = w_j \mathcal{I}_n.$$

That is, we get:

$$\sum_{j=1}^m \left\| \frac{\partial f_{w,b}(X)}{\partial X^{(j)}} \right\|_F = \sum_{j=1}^m \|w_j \mathcal{I}_n\|_F = \sum_{j=1}^m |w_j| \, \|\mathcal{I}_n\|_F = \|w\|_1 \sqrt{n}, \tag{4}$$

Finally, we get the equivalence of Deep Lasso to regular Lasso (with the adjustment of the Lasso penalty by $\sqrt{n}$):

$$\min_\theta \alpha \mathcal{L}_\theta(X, Y) + (1 - \alpha) \sum_{j=1}^m \left\| \frac{\partial f_{w,b}(X)}{\partial X^{(j)}} \right\|_F = \min_\theta \alpha \mathcal{L}_\theta(X, Y) + (1 - \alpha) \sqrt{n} \|w\|_1. \tag{5}$$

Additionally, we show the equivalence to First-Layer Lasso too. First-Layer Lasso applies a Group Lasso penalty to the weights of the first layer parameters:

$$\min_\theta \alpha \mathcal{L}_\theta(X, Y) + (1 - \alpha) \sum_{j=1}^m \|W^{(j)}\|_2,$$

where $W^{(j)}$ is the j-th column of weight matrix of the first hidden layer corresponding to the $j$-th feature. However, in the case of linear regression, the first layer output coincides with the model

Table 10: **Benchmarking feature selection methods for MLP and FT-Transformer downstream models on datasets with corrupted with Laplace noise extra features.**

| % | FS method + Model | AL | CH | EY | GE | HE | HI | HO | JA | MI | OT | YE | rank |
|---|---|---|---|---|---|---|---|---|---|---|---|---|---|
| 50 | No FS + MLP | 0.942 | -0.471 | 0.546 | 0.525 | 0.372 | 0.804 | -0.616 | 0.704 | -0.909 | 0.781 | -0.797 | 8.27 |
| 50 | Univariate + MLP | 0.955 | -0.448 | 0.532 | 0.531 | 0.345 | 0.810 | -0.627 | 0.717 | -0.920 | 0.791 | -0.827 | 7.00 |
| 50 | Lasso + MLP | 0.953 | -0.456 | 0.560 | 0.507 | 0.374 | 0.810 | -0.594 | 0.718 | -0.902 | 0.797 | -0.789 | 6.55 |
| 50 | 1L Lasso + MLP | 0.955 | -0.445 | **0.594** | 0.521 | 0.378 | **0.816** | -0.572 | **0.721** | -0.902 | 0.798 | **-0.776** | 3.09 |
| 50 | AGL + MLP | 0.955 | **-0.443** | 0.570 | 0.518 | 0.384 | 0.810 | -0.577 | 0.721 | -0.900 | **0.8** | -0.782 | 3.36 |
| 50 | LassoNet + MLP | 0.951 | -0.451 | 0.554 | 0.551 | **0.387** | 0.812 | **-0.55** | 0.716 | -0.904 | 0.797 | -0.778 | 4.45 |
| 50 | AM + MLP | 0.953 | -0.453 | **0.593** | 0.526 | **0.387** | 0.813 | -0.560 | 0.719 | -0.905 | 0.795 | -0.778 | 3.91 |
| 50 | RF + MLP | 0.953 | -0.449 | 0.563 | 0.558 | 0.379 | 0.810 | -0.583 | **0.723** | -0.905 | 0.789 | -0.787 | 5.00 |
| 50 | XGBoost + MLP | **0.956** | -0.454 | 0.570 | **0.6** | 0.385 | 0.813 | -0.560 | 0.720 | **-0.894** | **0.8** | -0.787 | 2.64 |
| 50 | Deep Lasso + MLP | **0.957** | **-0.441** | 0.589 | 0.539 | 0.383 | **0.816** | -0.556 | 0.720 | -0.900 | **0.801** | **-0.776** | **1.91** |
| 75 | No FS + MLP | 0.919 | -0.504 | 0.505 | 0.514 | 0.357 | 0.786 | -0.645 | 0.687 | -0.913 | 0.765 | -0.808 | 7.73 |
| 75 | Univariate + MLP | 0.950 | -0.612 | 0.515 | 0.493 | 0.338 | 0.739 | -0.621 | 0.688 | -0.921 | 0.779 | -0.838 | 7.64 |
| 75 | Lasso + MLP | 0.952 | -0.447 | 0.545 | 0.482 | 0.376 | 0.811 | -0.568 | 0.714 | -0.902 | 0.796 | -0.794 | 4.36 |
| 75 | 1L Lasso + MLP | 0.955 | -0.447 | 0.543 | 0.519 | 0.372 | **0.814** | -0.596 | 0.714 | -0.902 | 0.791 | **-0.781** | 3.73 |
| 75 | AGL + MLP | 0.954 | **-0.443** | 0.550 | 0.519 | **0.382** | 0.810 | -0.566 | **0.722** | -0.904 | 0.792 | -0.784 | 3.09 |
| 75 | LassoNet + MLP | 0.947 | -0.460 | 0.548 | 0.543 | 0.375 | 0.807 | -0.575 | 0.712 | -0.909 | 0.791 | -0.787 | 5.09 |
| 75 | RF + MLP | 0.951 | -0.450 | 0.564 | 0.524 | 0.367 | 0.805 | **-0.561** | 0.716 | -0.902 | 0.770 | -0.790 | 4.45 |
| 75 | XGBoost + MLP | **0.956** | -0.454 | **0.571** | **0.558** | 0.380 | 0.810 | **-0.555** | **0.721** | **-0.896** | **0.797** | -0.787 | **2.00** |
| 75 | Deep Lasso + MLP | 0.953 | **-0.443** | 0.552 | 0.527 | 0.377 | 0.812 | **-0.557** | **0.72** | -0.898 | **0.798** | **-0.78** | **2.00** |

output and has only a single logit. That is, the weight matrix $W$ is in fact a single row vector $w$ with $m$ entries giving us the equivalence between the First-Layer Lasso and regular Lasso, and therefore between First-Layer Lasso and Deep Lasso too in the case of linear regression:

$$\min_\theta \alpha \mathcal{L}_\theta(X, Y) + (1 - \alpha) \sum_{j=1}^{m} ||w^{(j)}||_2 = \min_\theta \alpha \mathcal{L}_\theta(X, Y) + (1 - \alpha) \|w\|_1.$$

☐

# F   More Results

Tables 16, 17, 18 are similar to the tables in the main body, but they include the standard deviations computed across seeds for each experiment. We also report performance of XGBoost baseline model for comparison, however we exclude it from rank computations.

## F.1   Corrupting features with Laplace noise

In addition to the experiments with Gaussian noise corruption, we conduct experiments with corrupting features with Laplace noise. In particular, we sample additional features from the original ones and corrupt them with Laplace noise with the standard deviation of original features:

$$x_c = 0.5x + 0.5\sigma\epsilon_l, \epsilon_l \sim Laplace(0, \frac{1}{\sqrt{2}})$$

We include results for this setup in Table 10. Similarly to the experiments with Gaussian noise corruption we observe that Deep Lasso along with XGBoost feature selection achieve the best downstream performance in terms of average rank.

## F.2   ROC-AUC, Precision and Recall Metrics

One common way of measuring performance of feature selection methods is using ROC-AUC, precision and recall scores, which indicate how often these methods choose "correct" and "incorrect" features. We instead focus on the downstream performance metric since in the case of corrupted and more prominently second-order extraneous features, it is not possible to treat feature selection as binary classification problems because additional features may contain useful information. However in this section we report precision, recall and ROC-AUC scores for the setup with random features, which can be reasonably treated as a binary classification problem. Tables 11 12 present ROC-AUC, and precision scores correspondingly with ranks computed taking importance into account. We note

Table 11: **ROC-AUC scores for feature selection methods trained on datasets with random extraneous features** Bold font indicates the best numbers dataset-wise and lower rank indicates better overall result.

| % | FS method | AL | CH | CO | EY | GE | HE | HI | HO | JA | MI | OT | YE | rank |
|---|---|---|---|---|---|---|---|---|---|---|---|---|---|---|
| 50 | Univariate | **1.0** | **1.0** | **1.0** | 0.916 | 0.926 | 0.982 | 0.866 | **1.0** | 0.706 | **0.993** | **1.0** | 0.990 | 2.50 |
| 50 | Lasso | 0.988 | **1.0** | 1.000 | 0.795 | 0.371 | 0.976 | 0.847 | 0.920 | 0.735 | 0.596 | 0.991 | 0.965 | 5.00 |
| 50 | 1L Lasso | 0.785 | **0.995** | 0.969 | 0.789 | 0.513 | 0.980 | 0.888 | 0.879 | 0.716 | 0.947 | 0.952 | 0.998 | 4.58 |
| 50 | AGL | 0.972 | 0.827 | 0.862 | 0.870 | 0.471 | **0.999** | 0.786 | **1.0** | **0.746** | 0.925 | 0.931 | 0.999 | 4.25 |
| 50 | LassoNet | **1.0** | **0.999** | **1.0** | 0.681 | 0.664 | 0.998 | 0.862 | 0.979 | 0.674 | 0.630 | 0.722 | 0.966 | 4.83 |
| 50 | AM | 0.848 | **0.992** | 0.752 | 0.765 | 0.770 | **1.0** | 0.876 | 0.980 | **0.759** | 0.704 | 0.972 | 0.988 | 4.33 |
| 50 | RF | 0.906 | **1.0** | 0.898 | 0.884 | **0.998** | **1.0** | 0.874 | 0.934 | 0.729 | 0.559 | 0.995 | 0.891 | 4.17 |
| 50 | XGBoost | 0.945 | **1.0** | 0.944 | **0.993** | 0.943 | **1.0** | 0.890 | **1.0** | **0.75** | 0.968 | 0.992 | 0.988 | 2.33 |
| 50 | Deep Lasso | **1.0** | **0.998** | 0.981 | 0.927 | 0.482 | **1.0** | **0.922** | **0.993** | 0.699 | 0.957 | **1.0** | **1.0** | 2.33 |
| 75 | Univariate | **1.0** | **1.0** | **1.0** | 0.918 | 0.872 | 0.986 | 0.846 | **1.0** | 0.731 | 0.993 | **1.0** | 0.991 | 2.08 |
| 75 | Lasso | **1.0** | **1.0** | 0.976 | 0.771 | 0.781 | 0.976 | 0.837 | 0.858 | **0.77** | 0.597 | 0.991 | 0.943 | 4.08 |
| 75 | 1L Lasso | 0.979 | **0.998** | 0.932 | 0.841 | 0.477 | 0.983 | 0.784 | 0.910 | 0.739 | 0.830 | 0.963 | **1.0** | 4.33 |
| 75 | AGL | 0.922 | **1.0** | 0.862 | 0.815 | 0.492 | 0.986 | 0.791 | 0.908 | 0.726 | 0.891 | 0.831 | 1.000 | 4.92 |
| 75 | LassoNet | **1.0** | **1.0** | 0.998 | 0.582 | 0.531 | 0.965 | 0.821 | 0.974 | 0.613 | 0.448 | 0.589 | 0.931 | 5.08 |
| 75 | Forest | 0.927 | **1.0** | 0.957 | 0.916 | **1.0** | **1.0** | **0.88** | 0.990 | 0.730 | 0.700 | 0.998 | 0.942 | 3.50 |
| 75 | XGBoost | 0.945 | **1.0** | 0.948 | **0.946** | **1.0** | **1.0** | 0.700 | 0.997 | 0.737 | **1.0** | 0.981 | 0.999 | 2.92 |
| 75 | Deep Lasso | 1.000 | **0.994** | 0.985 | 0.919 | 0.273 | **1.0** | **0.895** | 0.995 | **0.762** | 0.959 | 1.000 | **1.0** | 2.25 |

Table 12: **Precision scores for feature selection methods trained on datasets with random extraneous features** Bold font indicates the best numbers dataset-wise and lower rank indicates better overall result.

| % | FS method | AL | CH | CO | EY | GE | HE | HI | HO | JA | MI | OT | YE | rank |
|---|---|---|---|---|---|---|---|---|---|---|---|---|---|---|
| 50 | Univariate | **1.0** | **1.0** | **1.0** | 0.808 | 0.844 | 0.963 | 0.786 | **1.0** | 0.648 | **0.986** | **1.0** | 0.967 | 2.75 |
| 50 | Lasso | 0.987 | **1.0** | 0.989 | 0.704 | 0.403 | 0.963 | 0.750 | 0.875 | 0.661 | 0.588 | 0.969 | 0.922 | 5.00 |
| 50 | 1L Lasso | 0.776 | **0.988** | 0.954 | 0.719 | 0.497 | 0.937 | 0.811 | 0.856 | 0.644 | 0.882 | 0.915 | 0.988 | 4.67 |
| 50 | AGL | 0.946 | 0.738 | 0.794 | 0.777 | 0.469 | **0.989** | 0.704 | **1.0** | 0.665 | 0.870 | 0.875 | 0.987 | 4.50 |
| 50 | LassoNet | **1.0** | **0.988** | **1.0** | 0.619 | 0.625 | **0.985** | 0.771 | 0.931 | 0.644 | 0.582 | 0.692 | 0.908 | 5.00 |
| 50 | AM | 0.768 | **0.962** | 0.715 | 0.692 | 0.684 | **1.0** | 0.789 | 0.938 | 0.676 | 0.646 | 0.920 | 0.953 | 4.58 |
| 50 | RF | 0.901 | **1.0** | 0.893 | 0.885 | **0.984** | **1.0** | 0.807 | 0.875 | 0.661 | 0.553 | 0.985 | 0.809 | 4.25 |
| 50 | XGBoost | 0.945 | **1.0** | 0.944 | **0.969** | 0.894 | **1.0** | 0.818 | **1.0** | **0.981** | 0.956 | 0.992 | 0.977 | 2.17 |
| 50 | Deep Lasso | **1.0** | **0.988** | 0.974 | 0.835 | 0.488 | **1.0** | **0.861** | 0.969 | 0.641 | 0.950 | **1.0** | **1.0** | 2.67 |
| 75 | Univariate | **1.0** | **1.0** | **1.0** | 0.769 | 0.688 | 0.963 | 0.643 | **1.0** | 0.537 | 0.987 | **1.0** | 0.967 | 2.67 |
| 75 | Lasso | **1.0** | **1.0** | 0.957 | 0.608 | 0.766 | 0.963 | 0.679 | 0.812 | **0.622** | 0.401 | 0.963 | 0.878 | 3.75 |
| 75 | 1L Lasso | 0.882 | **0.975** | 0.909 | 0.665 | 0.231 | 0.948 | 0.632 | 0.881 | 0.531 | 0.721 | 0.933 | **0.996** | 5.17 |
| 75 | AGL | 0.856 | **1.0** | 0.743 | 0.685 | 0.253 | 0.952 | 0.654 | 0.869 | 0.541 | 0.775 | 0.704 | 0.987 | 4.75 |
| 75 | LassoNet | **1.0** | **1.0** | 0.976 | 0.458 | 0.488 | 0.919 | 0.650 | 0.912 | 0.415 | 0.331 | 0.524 | 0.770 | 5.08 |
| 75 | RF | 0.918 | **1.0** | 0.943 | **0.888** | **1.0** | **1.0** | 0.789 | 0.912 | 0.559 | 0.664 | 0.995 | 0.836 | 3.17 |
| 75 | XGBoost | 0.938 | **1.0** | 0.946 | **0.873** | **0.994** | **1.0** | 0.643 | 0.938 | 0.567 | **0.993** | 0.974 | 0.981 | 2.67 |
| 75 | Deep Lasso | 0.990 | **0.95** | 0.957 | 0.777 | 0.262 | **1.0** | **0.775** | 0.956 | 0.576 | 0.946 | 0.992 | **1.0** | 2.17 |

that in our setup the recall numbers are identical to the precision numbers, since we always "select" top-k features, where k is fixed. Therefore, the number of false positives is equal to the number of false negatives. We observe similar trends to results in the Table 1, where we evaluated algorithms with respect to the downstream performance. In particular, GBDT based approaches, univariate statistical test and Deep Lasso outperforming other considered feature selection methods.

## F.3 Does feature selection help bridging the gap with GBDT models?

In Section 4, we demonstrated that neural networks are more sensitive to noise than GBDT models, particularly the MLP architecture. In this section, we analyze whether feature selection helps narrow the gap between neural networks and GBDT models in the presence of noisy features. In Tables 13, 15, 14 we present results for the baseline XGBoost models, neural networks trained without feature selection, and neural networks trained with Deep Lasso feature selection, based on our benchmark. We observe that with feature selection, MLP models perform comparably to the XGBoost models in second-order setup and significantly better in random and corrupted features setup, while transformer models consistently outperform GBDT models in terms of average rank.

Table 13: **Benchmarking Neural Networks with feature selection against XGBoost models on datasets with random extra features.**

| % | FS method | AL | CH | CO | EY | GE | HE | HI | HO | JA | MI | OT | YE | rank |
|---|---|---|---|---|---|---|---|---|---|---|---|---|---|---|
| 50 | XGBoost Baseline | 0.858 | **-0.416** | 0.951 | **0.665** | **0.613** | 0.360 | 0.728 | -0.562 | **0.725** | **-0.885** | 0.801 | -0.797 | 1.83 |
| 50 | No FS + MLP | 0.941 | -0.480 | 0.961 | 0.538 | 0.466 | 0.366 | 0.798 | -0.622 | 0.703 | -0.911 | 0.773 | -0.801 | 2.58 |
| 50 | Deep Lasso + MLP | **0.959** | -0.443 | **0.968** | 0.573 | 0.485 | **0.383** | **0.814** | **-0.549** | 0.720 | -0.894 | **0.802** | **-0.776** | 1.42 |
| 75 | XGBoost Baseline | 0.839 | **-0.42** | 0.941 | **0.626** | **0.583** | 0.349 | 0.726 | **-0.57** | 0.718 | -0.894 | 0.796 | -0.802 | 1.92 |
| 75 | No FS + MLP | 0.925 | -0.527 | 0.955 | 0.502 | 0.417 | 0.348 | 0.778 | -0.674 | 0.671 | -0.917 | 0.749 | -0.812 | 2.67 |
| 75 | Deep Lasso + MLP | **0.957** | -0.446 | **0.969** | 0.569 | 0.479 | **0.387** | **0.814** | -0.559 | **0.721** | **-0.893** | **0.8** | **-0.774** | 1.25 |
| 50 | XGBoost Baseline | 0.858 | **-0.416** | 0.951 | 0.665 | **0.613** | 0.360 | 0.728 | **-0.562** | 0.725 | **-0.885** | 0.801 | -0.797 | 2.25 |
| 50 | No FS + FT | 0.959 | -0.432 | 0.966 | 0.673 | 0.500 | 0.384 | **0.817** | -0.577 | 0.730 | -0.902 | 0.813 | -0.792 | 2.08 |
| 50 | Deep Lasso + FT | **0.962** | **-0.419** | **0.969** | **0.703** | 0.504 | **0.392** | **0.817** | **-0.56** | **0.733** | -0.900 | **0.817** | **-0.788** | 1.17 |

Table 14: **Benchmarking Neural Networks with feature selection against XGBoost models on datasets with corrupted extra features.**

| % | FS method | AL | CH | CO | EY | GE | HE | HI | HO | JA | MI | OT | YE | rank |
|---|---|---|---|---|---|---|---|---|---|---|---|---|---|---|
| 50 | XGBoost Baseline | 0.854 | **-0.415** | 0.946 | **0.666** | **0.618** | 0.359 | 0.726 | **-0.559** | 0.723 | **-0.891** | 0.796 | -0.797 | 1.83 |
| 50 | No FS + MLP | 0.946 | -0.475 | 0.965 | 0.557 | 0.525 | 0.370 | 0.802 | -0.607 | 0.703 | -0.909 | 0.778 | -0.797 | 2.67 |
| 50 | Deep Lasso + MLP | **0.955** | -0.447 | **0.968** | 0.577 | 0.525 | **0.388** | **0.815** | -0.567 | 0.721 | -0.895 | **0.801** | **-0.776** | 1.58 |
| 75 | XGBoost Baseline | 0.830 | **-0.425** | 0.929 | **0.625** | **0.58** | 0.353 | 0.721 | -0.573 | **0.719** | **-0.894** | 0.795 | -0.801 | 1.92 |
| 75 | No FS + MLP | 0.921 | -0.516 | 0.956 | 0.518 | 0.503 | 0.356 | 0.788 | -0.632 | 0.686 | -0.913 | 0.762 | -0.808 | 2.67 |
| 75 | Deep Lasso + MLP | **0.959** | -0.441 | **0.968** | 0.554 | 0.517 | **0.386** | **0.813** | **-0.563** | 0.718 | -0.898 | **0.804** | **-0.778** | 1.42 |
| 50 | XGBoost Baseline | 0.854 | **-0.415** | 0.946 | 0.666 | **0.618** | 0.359 | 0.726 | **-0.559** | 0.723 | **-0.891** | 0.796 | -0.797 | 2.25 |
| 50 | No FS + FT | 0.960 | -0.430 | 0.967 | 0.686 | 0.576 | 0.386 | 0.818 | -0.574 | 0.731 | -0.901 | **0.809** | -0.793 | 2.17 |
| 50 | Deep Lasso + FT | **0.961** | -0.422 | **0.968** | **0.725** | 0.577 | **0.393** | **0.821** | **-0.561** | **0.736** | -0.898 | **0.809** | **-0.788** | 1.25 |

Table 15: **Benchmarking Neural Networks with feature selection against XGBoost models on datasets with secondorder extra features.**

| % | FS method | AL | CH | CO | EY | GE | HE | HI | HO | JA | MI | OT | YE | rank |
|---|---|---|---|---|---|---|---|---|---|---|---|---|---|---|
| 50 | XGBoost Baseline | 0.923 | **-0.429** | **0.969** | **0.709** | **0.683** | 0.374 | 0.729 | **-0.541** | 0.726 | **-0.846** | **0.825** | -0.781 | 1.58 |
| 50 | No FS + MLP | 0.960 | -0.443 | 0.969 | 0.631 | 0.605 | **0.383** | 0.811 | -0.549 | 0.719 | -0.891 | 0.800 | -0.786 | 2.42 |
| 50 | Deep Lasso + MLP | **0.961** | -0.441 | 0.969 | 0.648 | 0.600 | **0.384** | **0.815** | -0.572 | **0.733** | -0.890 | 0.805 | **-0.776** | 1.83 |
| 75 | XGBoost Baseline | 0.917 | **-0.414** | 0.968 | **0.74** | **0.691** | 0.376 | 0.731 | **-0.542** | 0.725 | **-0.849** | **0.825** | -0.789 | 1.67 |
| 75 | No FS + MLP | 0.952 | -0.451 | **0.969** | 0.630 | 0.598 | **0.388** | 0.808 | **-0.542** | 0.717 | -0.900 | 0.792 | -0.792 | 2.25 |
| 75 | Deep Lasso + MLP | **0.959** | -0.448 | **0.969** | 0.647 | 0.582 | 0.378 | **0.821** | -0.568 | **0.74** | -0.890 | 0.805 | **-0.78** | 1.67 |
| 50 | XGBoost Baseline | 0.923 | -0.429 | 0.969 | 0.709 | **0.683** | 0.374 | 0.729 | **-0.541** | 0.726 | **-0.846** | **0.825** | -0.781 | 2.00 |
| 50 | No FS + FT | 0.962 | -0.425 | 0.968 | **0.733** | 0.558 | **0.391** | 0.819 | -0.552 | 0.732 | -0.901 | 0.818 | -0.790 | 2.25 |
| 50 | Deep Lasso + FT | **0.963** | **-0.422** | **0.97** | 0.726 | 0.608 | 0.388 | **0.822** | -0.558 | **0.738** | -0.897 | 0.819 | -0.789 | 1.58 |

Table 16: **Benchmarking feature selection methods for MLP and FT-Transformer downstream models on datasets with random extra features.** We report performance of models trained on features selected by different FS algorithms in terms of accuracy for classification and negative RMSE for regression problems. % refers to percent of extra features in the dataset: either 50% or 75% features are second-order. Bold font indicates the best numbers dataset-wise and lower rank indicates better overall result.

| % | FS method | AL | CH | CO | EY | GE | HE | HI | HO | JA | MI | OT | YE | rank |
|---|---|---|---|---|---|---|---|---|---|---|---|---|---|---|
| 50 | XGBoost Baseline | 0.923±0.0 | -0.429±0.001 | 0.969±0.0 | 0.709±0.003 | 0.683±0.003 | 0.374±0.001 | 0.729±0.001 | -0.541±0.002 | 0.726±0.001 | -0.846±0.0 | 0.825±0.001 | -0.781±0.0 | |
| 50 | No FS + MLP | 0.941±0.001 | -0.48±0.007 | 0.961±0.001 | 0.538±0.005 | 0.466±0.007 | 0.366±0.001 | 0.798±0.001 | -0.622±0.007 | 0.703±0.001 | -0.911±0.002 | 0.773±0.004 | -0.801±0.001 | 8.08 |
| 50 | Univariate + MLP | **0.96±0.001** | -0.447±0.004 | 0.97±0.0 | 0.575±0.01 | 0.515±0.005 | 0.379±0.002 | 0.811±0.001 | **-0.549±0.008** | 0.715±0.002 | **-0.891±0.001** | **0.808±0.001** | -0.776±0.001 | 2.66 |
| 50 | Lasso + MLP | 0.949±0.001 | -0.454±0.004 | 0.969±0.001 | 0.547±0.016 | 0.458±0.018 | 0.38±0.002 | 0.812±0.0 | -0.599±0.003 | 0.715±0.003 | -0.907±0.001 | 0.805±0.004 | -0.787±0.001 | 5.91 |
| 50 | IL Lasso + MLP | 0.952±0.001 | -0.451±0.003 | 0.969±0.001 | 0.564±0.019 | 0.474±0.015 | 0.375±0.003 | 0.811±0.001 | -0.568±0.005 | 0.715±0.004 | -0.897±0.0 | 0.796±0.008 | **-0.773±0.001** | 4.91 |
| 50 | AGL + MLP | 0.958±0.001 | -0.512±0.064 | 0.969±0.0 | **0.578±0.017** | 0.473±0.018 | **0.386±0.003** | 0.81±0.001 | -0.557±0.003 | 0.718±0.001 | -0.898±0.001 | 0.799±0.005 | -0.778±0.001 | 4.33 |
| 50 | LassoNet + MLP | 0.954±0.001 | **0.445±0.004** | 0.969±0.001 | 0.552±0.006 | 0.495±0.01 | 0.385±0.002 | 0.811±0.003 | -0.557±0.008 | 0.715±0.001 | -0.907±0.001 | 0.783±0.003 | -0.787±0.002 | 5.16 |
| 50 | AM + MLP | 0.953±0.001 | **0.444±0.003** | 0.968±0.001 | 0.554±0.016 | 0.498±0.007 | 0.382±0.002 | **0.813±0.002** | -0.566±0.012 | **0.722±0.002** | -0.904±0.003 | 0.801±0.003 | -0.777±0.002 | 3.83 |
| 50 | RF + MLP | 0.955±0.001 | -0.453±0.003 | 0.969±0.0 | **0.589±0.011** | **0.594±0.023** | **0.386±0.002** | **0.814±0.001** | -0.572±0.006 | 0.72±0.001 | -0.904±0.0 | 0.806±0.002 | -0.786±0.001 | 2.91 |
| 50 | XGBoost + MLP | 0.956±0.001 | **0.444±0.002** | 0.969±0.0 | **0.59±0.015** | 0.502±0.013 | 0.385±0.002 | 0.812±0.0 | -0.56±0.014 | 0.72±0.003 | -0.893±0.001 | 0.805±0.003 | -0.777±0.001 | **2.33** |
| 50 | Deep Lasso + MLP | **0.959±0.001** | **0.443±0.002** | 0.968±0.001 | 0.573±0.007 | 0.485±0.006 | 0.383±0.002 | **0.814±0.0** | **-0.549±0.005** | 0.72±0.003 | -0.894±0.001 | 0.802±0.002 | -0.776±0.001 | **2.33** |
| 75 | XGBoost Baseline | 0.917±0.0 | -0.414±0.003 | 0.968±0.0 | 0.74±0.002 | 0.691±0.003 | 0.376±0.001 | 0.731±0.001 | -0.542±0.002 | 0.725±0.001 | -0.849±0.0 | 0.825±0.001 | -0.789±0.0 | |
| 75 | No FS + MLP | 0.925±0.002 | -0.527±0.006 | 0.955±0.001 | 0.502±0.007 | 0.417±0.005 | 0.348±0.002 | 0.778±0.002 | -0.674±0.007 | 0.671±0.002 | -0.917±0.002 | 0.749±0.003 | -0.812±0.001 | 7.41 |
| 75 | Univariate + MLP | **0.96±0.001** | **0.447±0.004** | 0.97±0.0 | 0.575±0.008 | 0.502±0.005 | 0.381±0.002 | 0.81±0.001 | **-0.549±0.008** | 0.713±0.001 | **-0.89±0.001** | **0.806±0.003** | -0.776±0.0 | 2.50 |
| 75 | Lasso + MLP | 0.959±0.001 | -0.454±0.002 | 0.967±0.0 | 0.543±0.013 | 0.491±0.009 | 0.381±0.002 | 0.811±0.001 | -0.612±0.004 | 0.716±0.002 | -0.907±0.001 | 0.802±0.003 | -0.789±0.001 | 4.33 |
| 75 | IL Lasso + MLP | 0.957±0.002 | **0.448±0.004** | 0.968±0.001 | 0.555±0.015 | 0.432±0.056 | 0.38±0.002 | 0.809±0.001 | -0.572±0.013 | 0.717±0.001 | -0.903±0.003 | 0.799±0.003 | **-0.775±0.002** | 4.41 |
| 75 | AGL + MLP | 0.954±0.002 | **0.447±0.003** | 0.968±0.001 | 0.561±0.013 | 0.429±0.03 | 0.382±0.002 | 0.809±0.005 | -0.571±0.012 | 0.719±0.003 | -0.901±0.0 | 0.762±0.035 | -0.777±0.001 | 4.33 |
| 75 | LassoNet + MLP | 0.958±0.001 | -0.452±0.006 | 0.966±0.002 | 0.528±0.009 | 0.475±0.005 | 0.383±0.004 | 0.809±0.003 | **-0.555±0.014** | 0.705±0.004 | -0.913±0.001 | 0.768±0.003 | -0.794±0.002 | 4.75 |
| 75 | RF + MLP | 0.949±0.001 | -0.453±0.003 | 0.968±0.001 | **0.584±0.008** | **0.61±0.008** | **0.386±0.002** | **0.814±0.001** | -0.585±0.012 | 0.718±0.002 | -0.902±0.0 | **0.808±0.003** | -0.784±0.001 | 2.91 |
| 75 | XGBoost + MLP | 0.958±0.001 | -0.451±0.003 | 0.969±0.0 | 0.576±0.007 | 0.583±0.015 | 0.382±0.002 | 0.81±0.0 | -0.568±0.009 | **0.72±0.002** | -0.892±0.0 | 0.804±0.002 | **-0.774±0.001** | **2.08** |
| 75 | Deep Lasso + MLP | 0.957±0.001 | **0.446±0.004** | 0.969±0.0 | 0.569±0.013 | 0.479±0.007 | **0.387±0.001** | **0.814±0.001** | **-0.559±0.015** | **0.721±0.002** | -0.893±0.001 | 0.8±0.002 | **-0.774±0.001** | 2.33 |
| 50 | No FS + FT | 0.959±0.002 | -0.432±0.002 | 0.966±0.001 | 0.673±0.015 | 0.5±0.004 | 0.384±0.002 | 0.817±0.001 | -0.577±0.005 | 0.73±0.002 | -0.902±0.004 | 0.813±0.003 | -0.792±0.002 | 6.58 |
| 50 | Univariate + FT | **0.963±0.001** | -0.424±0.002 | 0.97±0.001 | 0.7±0.009 | 0.519±0.01 | 0.389±0.002 | 0.819±0.001 | **-0.554±0.003** | 0.733±0.003 | **-0.897±0.004** | 0.819±0.001 | -0.789±0.001 | 2.83 |
| 50 | Lasso + FT | 0.952±0.003 | **0.419±0.003** | 0.96±0.01 | 0.682±0.018 | 0.489±0.013 | 0.388±0.003 | 0.819±0.0 | -0.594±0.011 | 0.728±0.002 | -0.999±0.0 | 0.817±0.0 | -0.998±0.0 | 6.33 |
| 50 | IL Lasso + FT | 0.963±0.001 | -0.423±0.004 | 0.969±0.0 | **0.72±0.011** | 0.489±0.012 | 0.382±0.003 | 0.818±0.001 | -0.577±0.008 | 0.732±0.003 | -0.904±0.004 | **0.819±0.002** | -0.791±0.003 | 5.16 |
| 50 | AGL + FT | 0.899±0.023 | **0.42±0.003** | 0.969±0.001 | 0.701±0.01 | 0.48±0.016 | 0.393±0.002 | **0.822±0.001** | -0.586±0.008 | 0.733±0.002 | -0.915±0.004 | 0.814±0.002 | -0.832±0.032 | 5.25 |
| 50 | LassoNet + FT | **0.963±0.001** | -0.426±0.004 | 0.97±0.001 | 0.67±0.01 | 0.505±0.005 | 0.392±0.002 | 0.818±0.002 | -0.559±0.005 | 0.733±0.002 | -0.904±0.004 | 0.808±0.002 | -0.791±0.001 | 4.83 |
| 50 | AM + FT | 0.962±0.001 | -0.425±0.003 | 0.968±0.001 | 0.657±0.047 | 0.505±0.01 | 0.389±0.002 | 0.821±0.001 | **-0.554±0.012** | 0.735±0.001 | -0.903±0.006 | 0.815±0.002 | **-0.789±0.001** | 3.75 |
| 50 | RF + FT | 0.962±0.001 | -0.42±0.003 | 0.969±0.001 | 0.718±0.014 | **0.591±0.011** | **0.395±0.002** | 0.821±0.001 | -0.558±0.007 | **0.737±0.001** | -0.9±0.004 | **0.82±0.001** | -0.791±0.001 | 1.75 |
| 50 | XGBoost + FT | **0.963±0.001** | -0.42±0.003 | 0.969±0.001 | **0.725±0.012** | 0.572±0.012 | 0.392±0.001 | 0.82±0.001 | -0.558±0.008 | 0.734±0.003 | **-0.898±0.005** | **0.82±0.002** | -0.789±0.001 | **1.91** |
| 50 | Deep Lasso + FT | 0.962±0.001 | **0.419±0.003** | 0.969±0.001 | 0.703±0.011 | 0.504±0.011 | 0.392±0.002 | 0.817±0.005 | -0.56±0.012 | 0.733±0.002 | -0.9±0.005 | 0.817±0.002 | **-0.788±0.002** | 3.66 |

Table 17: **Benchmarking feature selection methods for MLP and FT-Transformer downstream models on datasets with second-order extra features.** We report performance of models trained on features selected by different FS algorithms in terms of accuracy for classification and negative RMSE for regression problems. % refers to percent of extra features in the dataset: either 50% or 75% features are second-order. Bold font indicates the best numbers dataset-wise and lower rank indicates better overall result.

| % | FS method | AL | CH | CO | EY | GE | HE | HI | HO | JA | MI | OT | YE | rank |
|---|---|---|---|---|---|---|---|---|---|---|---|---|---|---|
| 50 | XGBoost Baseline | 0.854±0.0 | -0.415±0.001 | 0.946±0.0 | 0.666±0.003 | 0.618±0.003 | 0.359±0.001 | 0.726±0.001 | -0.559±0.002 | 0.723±0.001 | -0.891±0.001 | 0.796±0.001 | -0.797±0.0 | |
| 50 | No FS + MLP | 0.946±0.001 | -0.475±0.004 | 0.965±0.001 | 0.557±0.007 | 0.525±0.003 | 0.37±0.001 | 0.802±0.003 | -0.607±0.005 | 0.703±0.003 | -0.909±0.001 | 0.778±0.004 | -0.797±0.001 | 8.00 |
| 50 | Univariate + MLP | **0.955±0.001** | -0.451±0.004 | 0.966±0.001 | 0.556±0.007 | 0.514±0.005 | 0.346±0.003 | 0.81±0.001 | -0.62±0.004 | 0.717±0.002 | -0.92±0.0 | 0.795±0.001 | -0.828±0.001 | 7.33 |
| 50 | Lasso + MLP | **0.955±0.001** | **-0.449±0.003** | 0.968±0.001 | 0.548±0.017 | 0.512±0.011 | 0.382±0.002 | 0.813±0.002 | -0.602±0.002 | 0.713±0.001 | -0.903±0.0 | 0.796±0.001 | -0.795±0.001 | 5.42 |
| 50 | 1L Lasso + MLP | **0.955±0.002** | **-0.447±0.002** | 0.968±0.001 | 0.566±0.01 | 0.515±0.018 | 0.382±0.002 | 0.812±0.003 | -0.581±0.006 | 0.718±0.003 | -0.902±0.003 | 0.795±0.002 | -0.78±0.004 | 4.75 |
| 50 | AGL + MLP | 0.953±0.001 | -0.45±0.005 | 0.968±0.001 | **0.588±0.015** | 0.538±0.009 | 0.386±0.001 | 0.813±0.001 | -0.561±0.006 | 0.722±0.002 | -0.902±0.001 | 0.796±0.002 | -0.78±0.001 | 3.00 |
| 50 | LassoNet + MLP | **0.955±0.001** | -0.452±0.005 | **0.969±0.001** | 0.57±0.016 | 0.556±0.009 | 0.382±0.002 | 0.811±0.002 | **-0.551±0.007** | 0.719±0.002 | -0.905±0.001 | 0.795±0.004 | -0.777±0.001 | 3.83 |
| 50 | AM + MLP | **0.955±0.001** | **-0.449±0.003** | 0.967±0.001 | **0.583±0.012** | 0.527±0.009 | 0.381±0.002 | 0.814±0.001 | -0.555±0.012 | 0.722±0.001 | -0.905±0.004 | 0.797±0.003 | -0.78±0.002 | 3.58 |
| 50 | RF + MLP | 0.951±0.001 | -0.453±0.003 | 0.967±0.001 | 0.574±0.007 | **0.568±0.01** | 0.383±0.001 | 0.81±0.002 | -0.565±0.004 | **0.724±0.002** | -0.904±0.0 | 0.788±0.002 | -0.786±0.001 | 4.67 |
| 50 | XGBoost + MLP | 0.954±0.001 | -0.454±0.003 | **0.969±0.0** | **0.583±0.011** | 0.51±0.009 | 0.385±0.003 | **0.815±0.002** | -0.553±0.006 | 0.722±0.002 | **-0.892±0.001** | **0.803±0.002** | -0.779±0.001 | 2.67 |
| 50 | Deep Lasso + MLP | **0.955±0.001** | **-0.447±0.003** | 0.968±0.0 | 0.577±0.011 | 0.525±0.015 | **0.388±0.002** | **0.815±0.001** | -0.567±0.005 | 0.721±0.002 | -0.895±0.001 | 0.801±0.002 | **-0.776±0.001** | **2.58** |
| 75 | XGBoost Baseline | 0.83±0.0 | -0.425±0.001 | 0.929±0.0 | 0.625±0.003 | 0.58±0.002 | 0.353±0.001 | 0.721±0.002 | -0.573±0.001 | 0.719±0.001 | -0.894±0.001 | 0.795±0.001 | -0.801±0.0 | |
| 75 | No FS + MLP | 0.921±0.004 | -0.516±0.004 | 0.956±0.001 | 0.518±0.01 | 0.503±0.005 | 0.356±0.002 | 0.788±0.002 | -0.632±0.005 | 0.686±0.002 | -0.913±0.001 | 0.762±0.003 | -0.808±0.001 | 7.58 |
| 75 | Univariate + MLP | 0.955±0.0 | -0.569±0.004 | 0.941±0.001 | 0.51±0.008 | 0.495±0.005 | 0.347±0.002 | 0.742±0.001 | -0.62±0.004 | 0.686±0.001 | -0.921±0.0 | 0.779±0.002 | -0.838±0.001 | 7.50 |
| 75 | Lasso + MLP | 0.948±0.001 | -0.454±0.005 | 0.963±0.001 | **0.565±0.01** | 0.49±0.021 | 0.373±0.003 | 0.81±0.001 | -0.593±0.002 | 0.717±0.002 | -0.903±0.001 | 0.795±0.002 | -0.791±0.0 | 5.00 |
| 75 | 1L Lasso + MLP | 0.955±0.001 | -0.444±0.002 | 0.967±0.0 | 0.549±0.026 | 0.495±0.021 | 0.38±0.002 | 0.811±0.001 | -0.576±0.007 | **0.715±0.008** | -0.903±0.003 | 0.797±0.004 | -0.779±0.001 | 3.25 |
| 75 | AGL + MLP | 0.928±0.015 | -0.566±0.199 | 0.967±0.001 | 0.548±0.011 | 0.49±0.019 | 0.382±0.003 | 0.811±0.001 | -0.574±0.005 | 0.714±0.002 | -0.904±0.0 | 0.788±0.002 | -0.78±0.001 | 4.92 |
| 75 | LassoNet + MLP | 0.947±0.001 | -0.452±0.006 | **0.969±0.001** | 0.539±0.008 | **0.533±0.009** | 0.383±0.002 | 0.805±0.005 | -0.572±0.012 | 0.708±0.004 | -0.908±0.001 | 0.791±0.003 | -0.785±0.002 | 4.33 |
| 75 | RF + MLP | 0.952±0.001 | -0.45±0.003 | 0.963±0.001 | 0.547±0.011 | **0.533±0.007** | 0.372±0.003 | 0.805±0.002 | -0.573±0.005 | 0.716±0.002 | -0.903±0.0 | 0.765±0.003 | -0.788±0.001 | 4.92 |
| 75 | XGBoost + MLP | 0.954±0.001 | -0.515±0.048 | 0.968±0.001 | **0.571±0.009** | **0.53±0.01** | 0.381±0.002 | 0.811±0.001 | **-0.571±0.011** | **0.721±0.002** | -0.895±0.001 | 0.8±0.003 | -0.784±0.002 | 2.58 |
| 75 | Deep Lasso + MLP | **0.959±0.001** | **-0.441±0.002** | 0.968±0.0 | 0.554±0.015 | 0.517±0.006 | **0.386±0.002** | **0.813±0.002** | -0.563±0.009 | 0.718±0.003 | -0.898±0.001 | **0.804±0.003** | **-0.778±0.001** | **1.42** |
| 50 | No FS + FT | 0.96±0.001 | -0.43±0.004 | 0.967±0.001 | 0.686±0.008 | 0.576±0.022 | 0.386±0.002 | 0.818±0.001 | -0.574±0.007 | 0.731±0.002 | -0.901±0.003 | 0.809±0.002 | -0.793±0.001 | 6.08 |
| 50 | Univariate + FT | **0.963±0.001** | -0.422±0.003 | 0.965±0.001 | 0.681±0.011 | 0.574±0.013 | 0.345±0.002 | 0.812±0.002 | -0.628±0.007 | 0.733±0.001 | -0.92±0.002 | 0.812±0.002 | -0.826±0.0 | 6.17 |
| 50 | Lasso + FT | 0.952±0.004 | -0.422±0.003 | 0.936±0.037 | 0.697±0.022 | 0.556±0.012 | 0.387±0.002 | 0.82±0.0 | -0.586±0.006 | 0.732±0.003 | -0.937±0.0 | 0.812±0.002 | -0.915±0.004 | 6.42 |
| 50 | 1L Lasso + FT | 0.962±0.001 | **-0.419±0.002** | 0.969±0.001 | 0.718±0.008 | 0.571±0.009 | 0.389±0.002 | 0.82±0.001 | -0.57±0.004 | 0.731±0.001 | **-0.899±0.003** | **0.816±0.003** | -0.795±0.004 | 3.50 |
| 50 | AGL + FT | 0.906±0.021 | -0.426±0.003 | 0.969±0.001 | 0.697±0.018 | 0.591±0.009 | **0.392±0.002** | 0.82±0.001 | -0.552±0.005 | **0.735±0.002** | -0.914±0.005 | **0.816±0.003** | -0.83±0.016 | 4.08 |
| 50 | LassoNet + FT | **0.962±0.001** | -0.426±0.003 | 0.97±0.001 | 0.679±0.012 | 0.578±0.014 | **0.393±0.002** | 0.814±0.005 | -0.572±0.011 | **0.736±0.002** | -0.903±0.004 | 0.813±0.002 | **-0.79±0.001** | 3.92 |
| 50 | AM + FT | 0.962±0.001 | -0.424±0.002 | 0.969±0.0 | 0.68±0.017 | 0.572±0.018 | **0.392±0.001** | 0.82±0.001 | **-0.549±0.007** | 0.734±0.001 | -0.901±0.004 | **0.817±0.002** | -0.79±0.002 | 3.42 |
| 50 | RF + FT | 0.962±0.001 | -0.422±0.004 | 0.969±0.001 | 0.711±0.016 | 0.572±0.005 | 0.387±0.002 | 0.819±0.002 | -0.557±0.005 | 0.734±0.002 | **-0.898±0.002** | 0.806±0.002 | -0.793±0.002 | 3.42 |
| 50 | XGBoost + FT | **0.963±0.001** | -0.422±0.002 | **0.97±0.0** | 0.706±0.011 | 0.564±0.009 | 0.392±0.001 | **0.821±0.001** | **-0.548±0.003** | **0.735±0.001** | -0.897±0.003 | **0.816±0.001** | -0.79±0.001 | **2.42** |
| 50 | Deep Lasso + FT | 0.961±0.001 | -0.422±0.004 | 0.968±0.001 | **0.725±0.008** | 0.577±0.014 | **0.393±0.002** | **0.821±0.001** | -0.561±0.013 | 0.736±0.002 | -0.898±0.004 | 0.809±0.003 | **-0.788±0.001** | 2.67 |

Table 18: **Benchmarking feature selection methods for MLP and FT-Transformer downstream models on datasets with second-order extra features.** We report performance of models trained on features selected by different FS algorithms in terms of accuracy for classification and negative RMSE for regression problems. % refers to percent of extra features in the dataset: either 50% or 75% features are second-order. Bold font indicates the best numbers dataset-wise and lower rank indicates better overall result.

| % | FS method | AL | CH | CO | EY | GE | HE | HI | HO | JA | MI | OT | YE | rank |
|---|---|---|---|---|---|---|---|---|---|---|---|---|---|---|
| 50 | XGBoost Baseline | 0.923±0.0 | -0.429±0.001 | 0.969±0.0 | 0.709±0.003 | 0.683±0.003 | 0.374±0.001 | 0.729±0.001 | -0.541±0.001 | 0.726±0.001 | -0.846±0.0 | 0.825±0.001 | -0.781±0.0 | |
| 50 | No FS +MLP | 0.96±0.001 | -0.443±0.002 | 0.969±0.001 | 0.631±0.009 | 0.605±0.012 | **0.383±0.002** | 0.811±0.0 | **-0.549±0.006** | 0.719±0.002 | -0.891±0.002 | 0.8±0.003 | -0.786±0.001 | 4.50 |
| 50 | Univariate + MLP | **0.961±0.001** | -0.439±0.002 | 0.959±0.0 | 0.584±0.013 | 0.582±0.01 | 0.357±0.001 | 0.817±0.001 | -0.614±0.009 | 0.724±0.002 | -0.902±0.0 | 0.798±0.002 | -0.81±0.001 | 6.58 |
| 50 | Lasso + MLP | 0.955±0.001 | -0.443±0.004 | 0.966±0.004 | 0.608±0.012 | 0.59±0.022 | 0.366±0.007 | 0.816±0.001 | -0.564±0.004 | 0.724±0.002 | -0.891±0.0 | **0.806±0.001** | -0.783±0.0 | 5.33 |
| 50 | 1L Lasso + MLP | 0.959±0.0 | -0.445±0.005 | 0.969±0.001 | 0.634±0.014 | 0.571±0.036 | 0.38±0.003 | **0.815±0.007** | -0.565±0.01 | 0.728±0.001 | -0.89±0.0 | **0.808±0.003** | -0.78±0.001 | 3.92 |
| 50 | AGL + MLP | 0.961±0.001 | -0.443±0.008 | 0.953±0.052 | 0.637±0.015 | 0.594±0.019 | 0.383±0.002 | 0.807±0.007 | -0.565±0.007 | 0.73±0.009 | **-0.89±0.001** | **0.806±0.003** | **-0.776±0.001** | 3.25 |
| 50 | LassoNet + MLP | 0.959±0.001 | -0.442±0.004 | 0.969±0.0 | **0.641±0.011** | 0.611±0.006 | 0.379±0.001 | 0.816±0.002 | -0.595±0.01 | 0.724±0.003 | -0.893±0.0 | 0.797±0.002 | -0.784±0.002 | 4.50 |
| 50 | AM + MLP | **0.961±0.000** | **-0.439±0.005** | 0.968±0.001 | 0.622±0.009 | 0.604±0.01 | 0.381±0.003 | **0.819±0.001** | -0.566±0.033 | 0.73±0.002 | **-0.89±0.0** | 0.802±0.003 | -0.778±0.002 | 3.50 |
| 50 | RF + MLP | 0.958±0.001 | **-0.437±0.004** | 0.969±0.0 | 0.639±0.008 | **0.619±0.007** | 0.37±0.002 | 0.818±0.001 | -0.586±0.002 | **0.735±0.002** | -0.89±0.0 | 0.801±0.002 | -0.781±0.001 | 3.25 |
| 50 | XGBoost + MLP | 0.87±0.001 | -0.438±0.002 | **0.97±0.001** | 0.635±0.009 | 0.604±0.007 | 0.373±0.004 | **0.818±0.001** | -0.579±0.01 | **0.734±0.002** | -0.891±0.001 | 0.805±0.002 | -0.786±0.001 | 3.83 |
| 50 | Deep Lasso + MLP | **0.961±0.001** | **-0.441±0.007** | 0.969±0.001 | **0.648±0.009** | 0.6±0.01 | **0.384±0.001** | 0.815±0.001 | -0.572±0.017 | 0.733±0.001 | **-0.89±0.001** | 0.805±0.003 | **-0.776±0.001** | **2.67** |
| 75 | XGBoost Baseline | 0.917±0.0 | -0.414±0.003 | 0.968±0.0 | 0.74±0.002 | 0.691±0.003 | 0.376±0.001 | 0.731±0.001 | -0.542±0.002 | 0.725±0.001 | -0.849±0.0 | 0.825±0.001 | -0.789±0.0 | |
| 75 | No FS + MLP | 0.952±0.001 | -0.451±0.003 | **0.969±0.0** | 0.63±0.006 | **0.598±0.006** | **0.388±0.001** | 0.808±0.001 | **-0.542±0.006** | 0.717±0.002 | -0.9±0.001 | 0.792±0.003 | -0.792±0.002 | 4.92 |
| 75 | Univariate + MLP | 0.96±0.001 | -0.53±0.004 | 0.488±0.0 | 0.553±0.008 | 0.531±0.005 | 0.352±0.002 | 0.812±0.001 | -0.608±0.004 | 0.72±0.002 | -0.908±0.0 | 0.785±0.002 | -0.82±0.001 | 7.25 |
| 75 | Lasso + MLP | **0.96±0.001** | **-0.434±0.002** | **0.968±0.001** | 0.612±0.023 | 0.519±0.039 | 0.363±0.005 | 0.82±0.001 | -0.554±0.005 | 0.739±0.002 | -0.894±0.001 | **0.807±0.001** | -0.793±0.001 | 3.67 |
| 75 | 1L Lasso + MLP | **0.96±0.001** | -0.452±0.044 | 0.966±0.006 | **0.654±0.01** | 0.579±0.022 | 0.375±0.004 | 0.818±0.001 | -0.549±0.007 | **0.741±0.001** | -0.893±0.002 | 0.805±0.002 | -0.782±0.001 | 3.33 |
| 75 | AGL + MLP | 0.958±0.001 | -0.438±0.005 | 0.968±0.001 | **0.647±0.019** | 0.601±0.007 | 0.384±0.002 | 0.819±0.001 | **-0.545±0.007** | 0.736±0.001 | -0.893±0.0 | 0.8±0.003 | -0.781±0.001 | 2.67 |
| 75 | LassoNet + MLP | 0.958±0.001 | -0.454±0.027 | 0.968±0.001 | 0.633±0.015 | **0.615±0.012** | 0.362±0.002 | 0.813±0.005 | -0.569±0.011 | 0.726±0.002 | -0.895±0.0 | 0.793±0.003 | -0.786±0.002 | 4.83 |
| 75 | RF + MLP | 0.956±0.001 | -0.445±0.005 | 0.968±0.001 | 0.627±0.012 | 0.566±0.006 | 0.339±0.002 | 0.819±0.0 | -0.615±0.011 | 0.728±0.002 | -0.892±0.0 | 0.794±0.002 | -0.789±0.001 | 5.08 |
| 75 | XGBoost + MLP | 0.459±0.001 | -0.495±0.054 | 0.968±0.0 | 0.627±0.009 | 0.555±0.012 | 0.358±0.007 | **0.821±0.001** | -0.588±0.018 | 0.738±0.002 | -0.892±0.001 | 0.803±0.003 | -0.795±0.001 | 5.00 |
| 75 | Deep Lasso + MLP | 0.959±0.001 | -0.448±0.007 | **0.969±0.001** | **0.647±0.018** | 0.582±0.015 | 0.378±0.001 | **0.821±0.001** | -0.568±0.02 | **0.74±0.001** | **-0.89±0.0** | 0.805±0.002 | **-0.78±0.001** | **2.17** |
| 50 | No FS + FT | 0.962±0.001 | -0.425±0.003 | 0.968±0.001 | 0.733±0.011 | 0.558±0.011 | **0.391±0.002** | 0.819±0.001 | **-0.552±0.008** | 0.732±0.002 | -0.901±0.004 | 0.818±0.002 | -0.79±0.001 | 4.17 |
| 50 | Univariate + FT | **0.963±0.001** | **-0.419±0.003** | 0.942±0.002 | 0.615±0.009 | 0.573±0.011 | 0.351±0.002 | 0.82±0.0 | -0.613±0.005 | 0.728±0.001 | -0.91±0.003 | 0.81±0.001 | -0.819±0.001 | 6.67 |
| 50 | Lasso + FT | 0.962±0.002 | -0.422±0.003 | 0.967±0.004 | 0.72±0.011 | 0.56±0.015 | 0.381±0.003 | **0.823±0.0** | -0.561±0.011 | 0.733±0.002 | -0.905±0.002 | 0.816±0.002 | -0.874±0.002 | 5.25 |
| 50 | 1L Lasso + FT | **0.964±0.001** | -0.421±0.004 | **0.97±0.001** | **0.748±0.008** | **0.604±0.019** | **0.391±0.002** | 0.812±0.011 | -0.557±0.008 | 0.736±0.002 | -0.896±0.002 | 0.817±0.002 | -0.795±0.004 | 2.83 |
| 50 | AGL + FT | 0.935±0.024 | **-0.421±0.008** | 0.95±0.063 | **0.747±0.008** | 0.54±0.012 | 0.388±0.002 | 0.821±0.0 | **-0.557±0.008** | 0.73±0.008 | -0.907±0.004 | **0.821±0.002** | -0.821±0.014 | 4.17 |
| 50 | LassoNet + FT | **0.963±0.001** | -0.423±0.006 | **0.97±0.001** | 0.729±0.012 | **0.612±0.011** | 0.378±0.004 | 0.817±0.003 | -0.592±0.005 | 0.733±0.001 | -0.901±0.004 | 0.808±0.002 | -0.792±0.001 | 5.00 |
| 50 | AM + FT | 0.963±0.001 | **-0.418±0.002** | **0.97±0.001** | 0.719±0.009 | 0.597±0.019 | 0.387±0.003 | 0.82±0.001 | -0.579±0.024 | 0.736±0.002 | -0.902±0.005 | 0.819±0.002 | -0.791±0.002 | 3.83 |
| 50 | RF + FT | **0.963±0.001** | -0.422±0.006 | **0.97±0.001** | 0.733±0.007 | **0.615±0.008** | 0.365±0.008 | 0.822±0.001 | -0.6±0.013 | **0.738±0.001** | **-0.895±0.004** | 0.816±0.002 | -0.793±0.002 | 3.67 |
| 50 | XGBoost + FT | 0.871±0.001 | -0.424±0.003 | 0.962±0.001 | 0.737±0.014 | 0.594±0.013 | 0.379±0.002 | 0.822±0.001 | -0.587±0.014 | 0.736±0.001 | **-0.896±0.002** | 0.812±0.003 | -0.792±0.001 | 4.33 |
| 50 | Deep Lasso + FT | **0.963±0.001** | -0.422±0.008 | **0.97±0.001** | 0.726±0.011 | **0.608±0.012** | 0.388±0.002 | 0.822±0.001 | -0.558±0.005 | **0.738±0.001** | -0.897±0.003 | 0.819±0.001 | **-0.789±0.001** | **2.42** |

