# OpenReview forum: "A Performance-Driven Benchmark for Feature Selection in Tabular Deep Learning"
_NeurIPS.cc/2023/Track/Datasets_and_Benchmarks — NeurIPS 2023 Datasets and Benchmarks Poster_

### Official Review · Reviewer_rx9X · 2023-07-21
**A benchmark for feature selection methods in tabular deep learning**

**Rating:** 7
**Confidence:** 3
**Clarity:** The paper is well written, well struc…

**Strengths:**

1. The proposed benchmark takes into consideration different types of extraneous features, as well as including varied datasets in terms of number of samples and feature size, for both classification and regression tasks.
2. The feature selection methods used in the evaluation cover a wide range: statistical tests, tree model-based feature importance, attention-based and regularization coefficients.
3. The proposed Deep Lasso, although a more computationally demanding method compared to the other evaluated methods, provides a consistent improvement in scenarios considered to be "more challenging".

**Additional Feedback:**

I suggest investigating other types of extraneous features, such as spurious, as well as exploring other types of feature corruption, not limited to Gaussian noise.

**Correctness:**

The performance of the different feature selectors is evaluated in a sound way, by comparing the downstream performance using two different models. Additionally, measuring the statistical significance of the models performance differences in the method rank computation allows a fair comparison in the context of many scores differing by a small margin.

**Documentation:**

The benchmark is built on previously existing datasets, which are publicly available and widely-used. The authors provided the code for reproducibility as an appendix section. I encourage the authors to make it available in a code hosting, version control and collaboration platform.

**Ethics:**

There are no ethic concerns regarding the submissions. The benchmark utilizes publicly available tabular datasets.

**Limitations:**

The authors provide details on the limitation of the proposed algorithm from a computational perspective. I would add that the size of the benchmark might potentially be sub-optimal (12 datasets), as related work shows that there is no consistently outperforming method on tabular data, and model rankings might widely differ from one dataset to another. As tabular datasets are generally diverse, a larger collection of datasets might capture a broader range of their properties. I suggest the extension of the proposed benchmark to more datasets as a future work step.

**Opportunities For Improvement:**

1. The selected datasets contain only numerical features. An extension to a feature selection benchmark in datasets with a mixture of numerical and categorical features would provide a more realistic setup, given the heterogeneous nature of tabular data.
2. While it has been previously discussed that neural networks are more susceptible to noise than GBDT on small and medium-sized data (Grinsztajn et al., 2022), it would be interesting to provide an analysis of the proposed Deep Lasso method on different dataset size levels, and its potential impact on bridging the gap between NN and GBDT.

**Relation To Prior Work:**

The authors provided a detailed related work overview, structured in deep tabular learning methods, feature selection methods and existing benchmarks for feature selection.

**Summary And Contributions:**

The authors propose a benchmark for feature selection methods in tabular data learning on natural datasets, evaluating the robustness to scenarios involving corrupted, uninformative and redundant features. The quality of the selected features by each method is assessed as the downstream performance of MLP and FT-Transformer models using the selected features. Additionally, the authors propose Deep Lasso, a generalization of Lasso for deep neural networks and experimentally prove it outperforms several existing feature selection methods, especially in the more challenging scenarios.

---

> ### Author Response · Authors · 2023-08-23
> **Response to Reviewer rx9X**
>
> Thank you for your time and feedback. Inspired by your feedback, we have run several new experiments and updated our draft accordingly. We respond to each of your comments below:
>
> 1. _Including datasets with categorical features._
>
> Thank you for your suggestion! Unfortunately, we could not conduct experiments on additional datasets because of the prohibitive cost of these experiments (we include the details on computational cost of our experiments in Appendix C.1.). However, we will run these experiments for the camera-ready version. In particular, we plan to conduct experiments on the following datasets from the tabular benchmark introduced in [1,2]: Adult, Churn Modelling, Facebook Comments.
>
> 2. _It would be interesting to provide an analysis of the proposed Deep Lasso method on different dataset size levels, and its potential impact on bridging the gap between NN and GBDT._
>
> Prompted by your comment, we have now included results of the experiments comparing neural networks with Deep Lasso feature selection with the XGBoost model in Tables 13,14,15. We included a discussion of the impact of Deep Lasso for tabular neural networks on bridging the gap between NN and GBDT in Appendix F.3. Indeed, we find that with Deep Lasso, MLP models perform comparably to the XGBoost models in second-order setup and significantly better in random and corrupted features setup, while transformer models consistently outperform GBDT models across all setups.
>
> 3. _I encourage the authors to make it available in a code hosting, version control and collaboration platform._
>
> We decided to keep our submission anonymous, however we entirely agree with you and will release our code publicly on GitHub along with example notebooks after the review process concludes.
>
> 4. _I suggest investigating other types of extraneous features, such as spurious, as well as exploring other types of feature corruption, not limited to Gaussian noise._
>
> Thank you for your suggestion! To respond to your comment, we have conducted additional experiments involving a new feature corruption setup. Specifically, we select important features from extraneous features corrupted with Laplace noise. We evaluated 9 feature selection algorithms for the MLP downstream model, incorporating the new baselines suggested by other reviewers. The results of these experiments are aligned with our previous findings for selecting among features corrupted with Gaussian noise. Please, see the results of this experiment in Appendix F.1.
>
> In addition to the above, we have now conducted new experiments with two additional neural network baselines suggested by other reviewers, LassoNet[3] and Adaptive Group Lasso[4]. We include the results of these experiments in Tables 1,2,3. We find that both LassoNet and Adaptive Group Lasso perform similarly to classical Lasso and Group Lasso algorithms, while Deep Lasso and GBDT-based feature selection outperform other feature selection methods according to average ranks with Deep Lasso being more competitive in more difficult benchmark setups (such as corrupted features and second-order extraneous features).
>
> Thank you again for your thoughtful review. We added detailed discussions on our new experiments prompted by your review to Appendix F. We highlight revisions to our paper in blue color for easier navigation.
>
> We made a significant effort to address your questions, including new experiments and paper edits, and we would greatly appreciate it if you would consider raising your score in light of our response. Please let us know if you have additional questions we can address.
>
>
>
> [1] Gorishniy, Y., Rubachev, I., Khrulkov, V. and Babenko, A., 2021. Revisiting deep learning models for tabular data. Advances in Neural Information Processing Systems, 34, pp.18932-18943.
>
> [2] Gorishniy, Y., Rubachev, I. and Babenko, A., 2022. On embeddings for numerical features in tabular deep learning. Advances in Neural Information Processing Systems, 35, pp.24991-25004.
>
> [3] Lemhadri et al. 2021 Lassonet: A neural network with feature sparsity although cited has not been explored in the empirical results.
>
> [4] Dinh and Ho 2020 Consistent feature selection for analytic deep neural networks although cited has not been explored in the empirical results.

---

> > ### Comment · Reviewer_rx9X · 2023-08-23
> >
> > Thank you for including additional experiments based on my suggestions! I decided to raise my score to reflect the changes.

---

### Official Review · Reviewer_rBeS · 2023-07-29

**Rating:** 7
**Confidence:** 4
**Clarity:** Yes. The paper is well written and ea…

**Strengths:**


* **[Application Domain]** Tabular Datasets are very common in many practical situations (e.g., raw data from students, patients, financial records, and many more). While most modern ML methods are geared towards modality-rich data, we should encourage research on tabular datasets that are common in-practice.

* **[Comparison of Feature Selection methods]** The paper develops code to compare a number of feature selection methods. They releases their code. This allows future researchers to easily compare their own feature-selection method with any method implemented in the provided code.

* **[New Feature Selection Algorithm]**. Paper proposes a new method for feature selection. Specifically, a regularizer is added to the main (e.g., classification) objective that encourages the sum of L2 norm of columns of gradients to reduce. They mention that their method becomes equivalent to first-layer lasso in case the network is only 1 layer deep (though a theoretical proof is warranted, even if it was straight-forward)

**Additional Feedback:**

Are you able to run the experiments multiple times so that we get statistically-significant results? This will make your work more useful. If someone proposes a new feature selection method, all they have to do is run their new method multiple times and just copy-paste numbers from your paper.

**Correctness:**

The paper run a lot methods. They do good ablations (% of corrupted features VS choice of feature-selector VS model on datasets). However, the paper does not indicate the number of repetitions per experiment. In some cases, each repetition can yield different results (especially in gradient-based methods, such as, their proposed one and the 1st-layer-lasso).

**Documentation:**

Yes. Their code contains a README.md that explains how to re-run the reported baselines (it shows example bash commands)

**Ethics:**

No concerns

**Limitations:**

I dont see a limitations subsection.

**Opportunities For Improvement:**

* Lines 178 & 179 -- paper says Random features are more-often selected than Corrupted or higher-order features. However, Figure 2 shows the opposite.
* Line 184: The difference in the correlation is small. I wonder if the paper can comment on this.
* Line 216: Should be more than just a "note". I would expect to see some theorem & proof (even if proof is straight-forward). I also wonder about the similarity of Deep Lasso to first-layer Lasso.

**Relation To Prior Work:**

The existing methods that are plugged-into the benchmark are discussed appropriately.

**Summary And Contributions:**

Paper releases datasets for evaluating feature-selection methods on tabular datasets. Paper compares a number of classical and modern (Neural Nets) methods for feature selection. In addition, they propose their own feature selection method ("Deep Lasso"), which gives superior results to other methods.

---

> ### Author Response · Authors · 2023-08-23
> **Response to Reviewer rBeS**
>
> Thank you for your thoughtful review and encouraging feedback! We respond to each of your comments below:
>
> 1. _Typo in Lines 178 & 179_
>
> Thank you for pointing out the typo, we corrected it in our revised draft.
>
> 2. _Line 184: The difference in the correlation is small. I wonder if the paper can comment on this._
>
> Thanks for pointing this out.  Consider the random features case, where we synthesize extra non-predictive features with random noise.  In this case, selecting predictive features from random ones may be easy, but different selection algorithms assign different rankings within those two categories (e.g. they rank random features differently), leading to low rank correlations even in this setting where feature selection is straightforward. The problem of ranking predictive features amongst each other exists across setups and thus leads to small differences in rank correlation across setups.  We have updated our draft with a clarification.
>
> 3. _Line 216: Should be more than just a "note". I would expect to see some theorem & proof (even if proof is straight-forward). I also wonder about the similarity of Deep Lasso to first-layer Lasso._
>
> Thank you for your suggestion! We have now included a formal proof of equivalence in the linear regression case between classical Lasso, First-Layer Lasso, and Deep Lasso applied to model output in Appendix Section E. We have also referenced this proof in the main body.
>
> 4. _I don’t see a limitations subsection._
>
> We would like to point out that we have a limitations section in Appendix Section A.
>
> 5. _The paper does not indicate the number of repetitions per experiment. In some cases, each repetition can yield different results._
>
> We run each experiment with 10 random model initializations (seeds) after we find the optimal hyperparameters for feature selection and downstream models. We report average test metrics as well as model ranks. When calculating the average ranks, we consider the statistical significance of performance differences among the models. Please, see the details on the number of seeds and rank calculations in Appendix Section C.4.
>
>
> In addition to the above, we have now conducted new experiments with two additional neural network baselines suggested by other reviewers, LassoNet[1] and Adaptive Group Lasso[2]. We include the results of these experiments in Tables 1,2,3. We find that both LassoNet and Adaptive Group Lasso perform similarly to classical Lasso and Group Lasso algorithms, while Deep Lasso and GBDT-based feature selection outperform other feature selection methods according to average ranks with Deep Lasso being more competitive in more difficult benchmark setups (such as corrupted features and second-order extraneous features).
>
> We also conducted additional experiments involving a new feature corruption setup. Specifically, we select important features from extraneous features corrupted with Laplace noise. We evaluated 9 feature selection algorithms for the MLP downstream model, incorporating the new baselines suggested by other reviewers. The results of these experiments are aligned with our previous findings for selecting among features corrupted with Gaussian noise.
>
> We highlight revisions to our paper in blue color for easier navigation. Thank you again for your review. We hope we were able to address your comments. Please let us know if you have additional questions we can address.
>
> [1] Lemhadri et al. 2021 Lassonet: A neural network with feature sparsity although cited has not been explored in the empirical results.
>
> [2] Dinh and Ho 2020 Consistent feature selection for analytic deep neural networks although cited has not been explored in the empirical results.

---

### Official Review · Reviewer_Lpp4 · 2023-08-02
**Novelty component and experimental evidence need improvement.**

**Rating:** 5
**Confidence:** 3
**Clarity:** The writing of the paper is fair.

**Strengths:**

The authors go beyond the standard Gaussian error variable to assess the importance of variable selection. The proposition of other sources of erroneous variables opens the avenue for describing nuisance variables. Also the deep lasso idea is an interesting approach to design variable importance.

**Additional Feedback:**

NA

**Correctness:**

The claims of the paper are not fully correct. Versions of deep Lasso have already been explored in the literature thus not a completely novel contribution.

 The standard errors of the results have not been reported which limits the opportunity to understand variability in the results.

The baseline approaches chosen are very naive and do not cover majority of the new approaches available in the literature.

**Documentation:**

Yes there is sufficient detail on  data collection and organization, availability and maintenance, and ethical and responsible use.

**Limitations:**

There are no negative societal impacts.

**Opportunities For Improvement:**

A Bayesian version of deep Lasso has already been explored in the work  Liu 2021 "Variable Selection with Rigorous Uncertainty Quantification using Deep Bayesian Neural Networks: Posterior Concentration and Bernstein-von Mises Phenomenon"   and the work must be cited. The claim of the authors that deep Lasso is their new contribution becomes a bit questionable.

Although the authors suggest three ways of defining nuisance variables, it still limits the number of ways in which outliers are defined. Also ROC curves, precision and recall are one of the best ways to analyze the performance of a variable selection approach. Neither of these metrics have been explored in this work.

Why for transformer models only AM is explored but 1L Lasso dropped. Why for MLP models 1L Lasso is explored but AM dropped?

**Relation To Prior Work:**

Many prior works have not been cited or used (see also Opportunities for improvement)

Another comparative baseline Lemhadri  et al. 2021 Lassonet: A neural network with feature sparsity although cited has not been explored in the empirical results.

Another comparative baseline Dinh and Ho 2020 Consistent feature selection for analytic deep neural networks  although cited has not been explored in the empirical results.

**Summary And Contributions:**

The authors provide a challenging feature selection benchmark and evaluate the performance of various methods with respect to this benchmark. The authors propose an input-gradient-based analogue of Lasso for neural networks which outperform on the feature selection benchmark proposed by them.

---

> ### Author Response · Authors · 2023-08-23
> **Response to Reviewer Lpp4 (Part 1)**
>
> Thank you for your time and feedback. Inspired by your feedback, we have run several new experiments and updated our draft accordingly. We respond to each of your comments below:
>
> 1. _A Bayesian version of Deep Lasso_
>
> Thank you for bringing to our attention this interesting work, and we have now discussed and cited it in our updated draft.  We would like to point out significant differences between the proposed method and Deep Lasso.  The credible interval method uses the BNN posterior, estimated using HMC which is a computationally expensive algorithm, hence why the work in question only uses a 2-layer MLP with only 50 hidden units in their experiments and examines only small datasets with relatively few features (<=200).  In contrast, Deep Lasso minimizes the gradient norm during model training in order to select features and is scalable to state-of-the-art architectures and large datasets on limited computational resources.  The credible interval method in this paper also uses the 2-norm rather than the 1-norm.  On the one hand, the credible interval method is principled and comes with theoretical guarantees, but on the other hand Deep Lasso is scalable, easy to implement, and performs well empirically.  Finally, we want to point out that our work is a benchmark rather than a method paper, and Deep Lasso is an extension of Lasso that serves as a strong neural network baseline.  We agree that the paper you suggested is interesting and worth consideration, and we now compare it to our work in our draft.
>
> 2. _Although the authors suggest three ways of defining nuisance variables, it still limits the number of ways in which outliers are defined._
>
> Prompted by your comment, we conducted additional experiments involving a new feature corruption setup. Specifically, we select important features from extraneous features corrupted with Laplace noise. We evaluated 9 feature selection algorithms for the MLP downstream model, incorporating the new baselines you recommended. The results of these experiments are aligned with our previous findings for selecting among features corrupted with Gaussian noise. Please, see the results of this experiment in Appendix F.1.  Nonetheless, we agree that there are numerous kinds of feature selection scenarios that could occur in the real world, and we hope that the community will contribute new feature selection settings to our evolving open-source benchmark.
>
> 3. _ROC curves, precision and recall metrics_
>
> Initially, we omitted precision and recall metrics because, in benchmark setups involving corrupted and more prominently second-order features, we cannot consider extraneous features as inherently "unimportant.” This is because some corrupted or second-order features might offer more utility to the downstream neural network model than the "original" ones. Corruption could act as a regularization method and prevent overfitting, while second-order features could capture non-linear relationships between features. As a result, in these setups, we cannot treat feature selection as a binary classification problem. Instead, we focused on the performance of the downstream model as our primary metric. However, in response to your comment, we have now included precision, recall, and ROC-AUC scores for the scenario involving random features. In this case, we can treat feature selection as a binary classification problem. For more details, please refer to our results in the Appendix F.2.
>
> 4. _Why for transformer models only AM is explored but 1L Lasso dropped. Why for MLP models 1L Lasso is explored but AM dropped?_
>
> Originally, we utilized these methods specifically for FT-Transformer and MLP downstream models mainly because we conducted experiments with feature selection methods based on the same architecture as the downstream model. However, we agree that mixing and matching is interesting too, and so we have now included results for AM feature selection with downstream MLPs and 1L Lasso for downstream FT-Transformer. Please, see the results in Tables [1,2,3].

---

> ### Author Response · Authors · 2023-08-23
> **Response to Reviewer Lpp4 (Part 2)**
>
> 5. _The standard errors of the results have not been reported which limits the opportunity to understand variability in the results._
>
> We would like to point out that standard errors of all our results were included in the Appendix Tables [16,17,18], which contain a full slate of our results. We have added a clarification on that in the main body to our revision.
>
> 6. _Additional baselines_
>
> Thank you for referencing additional feature selection baselines. Prompted by your comment, we have now conducted additional experiments with LassoNet [1] and Adaptive Group Lasso [2] on our feature selection benchmark. We include the results of these experiments in Tables [1,2,3]. We find that both Lassonet and Adaptive Group Lasso perform similarly to classical Lasso and Group Lasso algorithms, while Deep Lasso and GBDT-based feature selection outperform other feature selection methods according to average ranks with Deep Lasso being more competitive in more difficult benchmark setups (such as corrupted features and second-order extraneous features).
>
> Thank you again for your thoughtful review. We added detailed discussions on our new experiments prompted by your review to the main body and to Appendix F. We highlight revisions to our paper in blue color for easier navigation. We made a significant effort to address each of your questions, including new experiments and paper edits, and would appreciate it if you would consider raising your score in light of our response. Please let us know if you have additional questions we can address.
>
> [1] Lemhadri et al. 2021 Lassonet: A neural network with feature sparsity although cited has not been explored in the empirical results.
>
> [2] Dinh and Ho 2020 Consistent feature selection for analytic deep neural networks although cited has not been explored in the empirical results.

---

> ### Author Response · Authors · 2023-08-28
> **Following up with Reviewer Lpp4**
>
> Thank you again for your thoughtful review. Does our response address your feedback? If you have any remaining concerns, we would appreciate an opportunity to engage further.

---

> > ### Comment · Reviewer_Lpp4 · 2023-08-28
> > **Response to review**
> >
> > I'm satisfied with responses and increased the score by 1.

---

### Decision · Program_Chairs · 2023-09-22

**Decision:**

Accept (Poster)

**Comment:**

Most structured databases tabular data. Feature selection is of significance to the ML community (e.g., for compute efficiency and reducing overfitting). Paper releases a dataset of tabular data. In addition, they propose a Deep Lasso Regularizer which can be added to the objective function for training deep networks, proving its equivalence to classical Lasso when the network is 1 layer deep.

Overall, this is a nice contribution of a dataset and a feature selection algorithm.